# Instruct2See: Learning to Remove Any Obstructions Across Distributions

**Junhang Li** [1 2 *]   **Yu Guo** [2 3 *]   **Chuhua Xian** [1]   **Shengfeng He** [2]

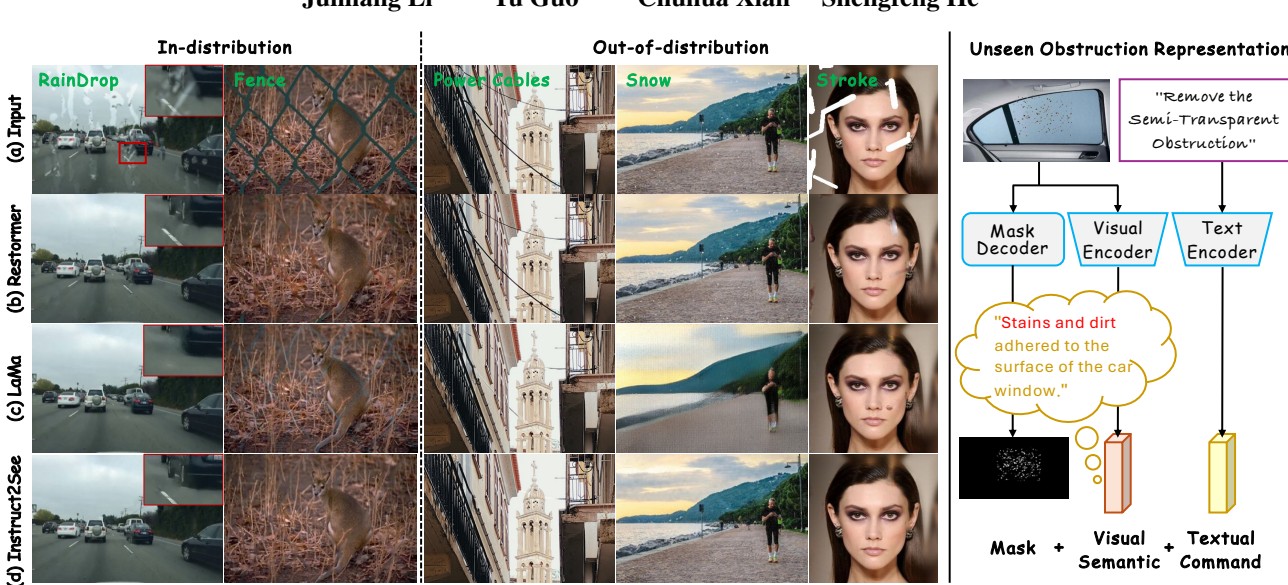

Figure 1: We present Instruct2See, a zero-shot framework for obstruction removal that handles arbitrary obstructions. It effectively tackles both soft (semi-transparent) and hard (opaque) obstructions, while demonstrating a robust effect in both in-distribution (seen) and out-of-distribution (unseen) scenes. The right part shows how we represent unseen obstructions.

## Abstract

Images are often obstructed by various obstacles due to capture limitations, hindering the observation of objects of interest. Most existing methods address occlusions from specific elements like fences or raindrops, but are constrained by the wide range of real-world obstructions, making comprehensive data collection impractical. To overcome these challenges, we propose Instruct2See, a novel zero-shot framework capable of handling both seen and unseen obstacles. The core idea of our approach is to unify obstruction removal by treating it as a soft-hard mask restoration problem, where any obstruction can be represented using multi-modal prompts, such as visual semantics and textual instructions, processed through a cross-attention unit to enhance contextual understanding and improve mode control. Additionally, a tunable mask adapter allows for dynamic soft masking, enabling real-time adjustment of inaccurate masks. Extensive experiments on both in-distribution and out-of-distribution obstacles show that Instruct2See consistently achieves strong performance and generalization in obstruction removal, regardless of whether the obstacles were present during the training phase. Code and dataset are available at https://jhscut.github.io/Instruct2See.

*Equal contribution  [1]School of Computer Science and Engineering, South China University of Technology [2]School of Computing and Information Systems, Singapore Management University [3]School of Navigation, Wuhan University of Technology. Correspondence to: Chuhua Xian and Shengfeng He <chhxian@scut.edu.cn, shengfenghe@smu.edu.sg>.

*Proceedings of the 42nd International Conference on Machine Learning*, Vancouver, Canada. PMLR 267, 2025. Copyright 2025 by the author(s).

## 1. Introduction

Obstruction removal is a challenging task that involves recovering clean scenes occluded by unwanted obstacles or unpredictable natural phenomena. Existing methods often focus on specific types of obstructions (Zhang et al., 2019; Wen et al., 2019; Du et al., 2020; Chugunov et al., 2024; Quan et al., 2023a; Huang et al., 2024b; Zhou et al., 2023; Li et al., 2024) by relying on predefined categories and specific training datasets. However, this reliance limits their generalization, often resulting in poor or invalid removal of occluders outside the training distribution. Therefore, it

is crucial to enable models to grasp the underlying physical properties of occlusion to enhance image quality under complex and varied conditions.

While many advanced methods (Tsogkas et al., 2023; Chugunov et al., 2024; Zhou et al., 2023; Dai et al., 2023; Zhang et al., 2023a; Li et al., 2024; Chang et al., 2024; Chen et al., 2023a) continue to target specific obstructions, the diversity of real-world obstacles makes designing and training separate models for each type inefficient and impractical (Guo et al., 2024). As a result, there is growing interest in all-in-one restoration models (Li et al., 2020; Chen et al., 2021; Han et al., 2022; Wang et al., 2023b; Valanarasu et al., 2022; Li et al., 2022a; Özdenizci & Legenstein, 2023), which aim to handle multiple complex scenarios with a single model. However, despite these advancements, such models remain constrained by their training datasets. As shown in Fig. 1 (b), restoration methods trained on multi-scene datasets struggle to handle unseen scenarios and lack the flexibility to adapt based on user input. This limitation is especially problematic in dynamic real-world applications, such as autonomous driving and intelligent robotics.

An alternative approach to obstruction removal is image inpainting techniques (Zeng et al., 2019; Suvorov et al., 2022), which repair or fill in missing or occluded regions by generating plausible pixel values that blend with the original scene. While inpainting can produce visually convincing results, these reconstructions often lack realism. For example, as shown in Fig. 1 (c), inpainting can fill occluded areas, but the reconstructed textures, such as the woman's face, often appear unnatural. Applying these inaccurate results to downstream tasks, like object detection, depth estimation, or video analysis, can lead to errors and negatively impact practical applications.

In this work, we revisit the problem of obstruction removal through the lens of unified masking and introduce Instruct2See, a method that transcends traditional training-dependent solutions by generalizing beyond specific data distributions (Fig. 1 (d)). Our distribution-agnostic approach formulates obstruction removal as a soft-hard mask restoration problem, where any obstruction can be represented by integrating visual semantic embeddings and text instructions, as provided by a visual-language model (right part of Fig. 1). By seamlessly integrating obstruction positions, visual semantics, and textual instructions, our approach redefines obstruction removal as an adaptable process that fluidly transitions between hard and soft masking, effectively capturing the complexity and diversity of real-world obscured scenarios. To be specific, visual semantics help recover missing information caused by occlusions, leading to more accurate scene reconstruction, while the text instruction serves as a prior prompt to guide various removal tasks. Additionally, we design a tunable mask adapter to

bridge the gap between the estimated and actual masks, reweighting the predicted mask into a soft mask that dynamically adapts to the testing scene. In summary, the key contributions of our work include:

- We introduce the first unified obstruction formulation and a novel zero-shot paradigm capable of handling any obstruction by integrating obstacle positions with multi-modal prompts, including visual semantics and text instructions.

- We develop a dynamic soft masking strategy that automatically refines inaccurate masks for occluding obstructions using a tunable adapter.

- Comprehensive experiments demonstrate the superior effectiveness of our model in obstruction removal, as well as its strong zero-shot generalization to unseen obstructions outside the training distribution.

## 2. Related Work

**Obstruction Removal.** The task of obstruction removal aims to clear unwanted obstructions from a scene, improving its visibility. Many existing methods are tailored to specific types of degradation, such as deraining (Zhang et al., 2023b; Wang et al., 2023a), desnowing (Quan et al., 2023b; Chen et al., 2023b), and raindrop removal (Qian et al., 2018; Li et al., 2024). While these approaches are effective for individual obstructions, they struggle with handling multiple degradation types simultaneously, often requiring separate models for each. To address this limitation, all-in-one image restoration models have been developed. For example, Liu et al. (2021) proposed a method that separates an image into obstruction and background layers using layered decomposition, improving visibility through obstructions. Li et al. (2022a) introduced AirNet, which incorporates an additional encoder with contrastive learning to distinguish between various degradation types. Potlapalli et al. (2023) presented PromptIR, a flexible plugin module that uses lightweight prompts to handle multiple image restoration tasks. Histoformer was introduced to employ histogram equalization techniques within a neural framework to enhance and restore degraded images (Sun et al., 2024). Despite these advancements, all-in-one models still face challenges when dealing with degradation types beyond their training data, limiting their effectiveness in real-world applications.

**Image Inpainting.** With advancements in parallel computing and deep learning, numerous image inpainting methods have been developed to restore missing or damaged regions in digital images with natural and coherent content. CNN-based methods, such as PEN-Net (Zeng et al., 2019) and LaMa (Suvorov et al., 2022), have proven efficient for generating local textures, but they often struggle to capture global context and handle complex patterns. To address

these limitations, transformer-based and diffusion-based models have been proposed. Deng et al. (2021) introduced the Contextual Transformer Network, which uses multi-scale, multi-head attention to capture long-range dependencies and global context through self-attention. Similarly, Li et al. (2022c) developed the Mask-Aware Transformer, which selectively aggregates non-local information using a dynamic mask, ensuring high fidelity and diversity in restored images. Diffusion-based methods like RePaint (Lugmayr et al., 2022) combine denoising diffusion probabilistic models with conditional inpainting to iteratively generate high-quality results. More recently, Grechka et al. (2024) proposed GradPaint, a diffusion-based method that leverages gradient guidance to enhance the quality and coherence of inpainted regions, producing artifact-free restorations.

Despite these efforts, applying image inpainting methods directly to obstruction removal is challenging due to limitations in cross-domain applicability and the distinct focus of these methods. While inpainting aims to generate visually plausible results, obstruction removal requires precise restoration to ensure data integrity for further analysis. Unrealistic outcomes can lead to errors in downstream tasks. Nonetheless, rethinking obstruction removal from the perspective of image inpainting presents a promising avenue for future exploration.

**Vision-Language Models (VLMs).** VLMs (Zhang et al., 2024) have gained significant attention for their ability to jointly interpret visual and textual information. Pre-trained VLMs, such as CLIP (Radford et al., 2021), have demonstrated improved performance across a range of downstream tasks by integrating visual and textual representations. CLIP employs a contrastive learning approach to align image and text embeddings, while distancing mismatched pairs in the embedding space. This alignment enables CLIP to perform zero-shot learning, recognizing unseen objects and concepts based on textual descriptions, achieving remarkable results without task-specific fine-tuning. In this work, we integrate the pre-trained CLIP with a prompt module to effectively leverage contextual information about degradation types, enhancing the performance of obstruction removal.

**Zero-Shot Learning (ZSL).** ZSL is an advanced machine learning paradigm that enables models to recognize and understand instances they have never encountered during training. Unlike traditional supervised learning, which relies on labeled examples for each class, ZSL leverages auxiliary information such as semantic attributes, textual descriptions, or word embeddings to generalize from seen to unseen classes. ZSL has shown significant potential in various applications, including image classification (Naeem et al., 2024), object detection (Huang et al., 2024a), and object counting (Zhu et al., 2024), offering a solution for scenarios with limited labeled data or dynamic class distributions. In

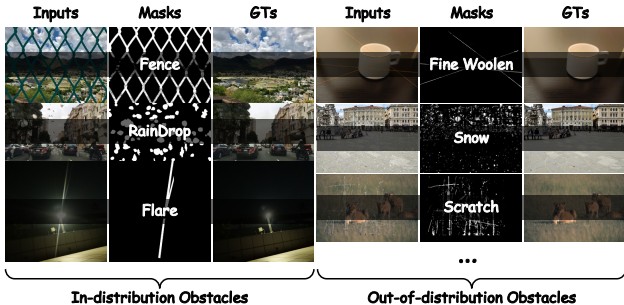

Figure 2: Examples of in- and out-of-distribution obstacles.

obstruction removal, characterized by diverse and varied obstructions, a ZSL approach is essential for handling a wide range of unseen obstacles effectively.

## 3. Distribution-Agnostic Formulation

**Seen Obstructions.** As illustrated in Fig. 2, we focus on three typical but distinct types of obstructions in the training data: *fences*, *raindrops*, and *flares*. These obstructions were chosen for their diversity in visual characteristics and mask extraction difficulty. *Fences* represent obstructions with a sharp distinction from the background, making it relatively straightforward to extract the mask. Therefore, we adopt a *hard masking* strategy for fences, as shown by the purple process in Fig. 3. In contrast, *raindrops* pose a challenge due to their blurred boundaries with the background and their random distribution across the image. Similarly, *flares* also exhibit soft, blurred edges, but their occurrence is more predictable, often appearing around point light sources. Given the difficulty in extracting accurate masks for raindrops and flares, which have indistinct boundaries, we employ a *soft masking* approach, indicated by the green process in Fig. 3.

**Unseen Obstructions.** In addition to the seen obstructions present in the training data, this work also targets more complex and varied unseen obstructions, including *power cables*, *yarn*, *snow*, *rain streaks*, *scratches*, and others, with examples depicted in Fig. 2. Depending on the degree of boundary ambiguity between the obstruction and the background, and the difficulty in extracting an appropriate mask, we apply *soft masking* for semi-transparent occlusions like *shadow* and *rain streaks*, where the edges are blurred. For more opaque occlusions, such as *power cables*, *yarn*, and *scratches*, we employ *hard masking* due to the clearer boundary between the obstruction and the background.

**Unified Imaging Description.** The overarching objective of this work is to remove unwanted obstructions and restore the occluded background. This problem can be modeled mathematically as follows:

$$I(x) = B(x) \circ (1 - M(x)) + R(x) \circ M(x), \quad (1)$$

where $I(x)$ represents the input image containing obstruc-

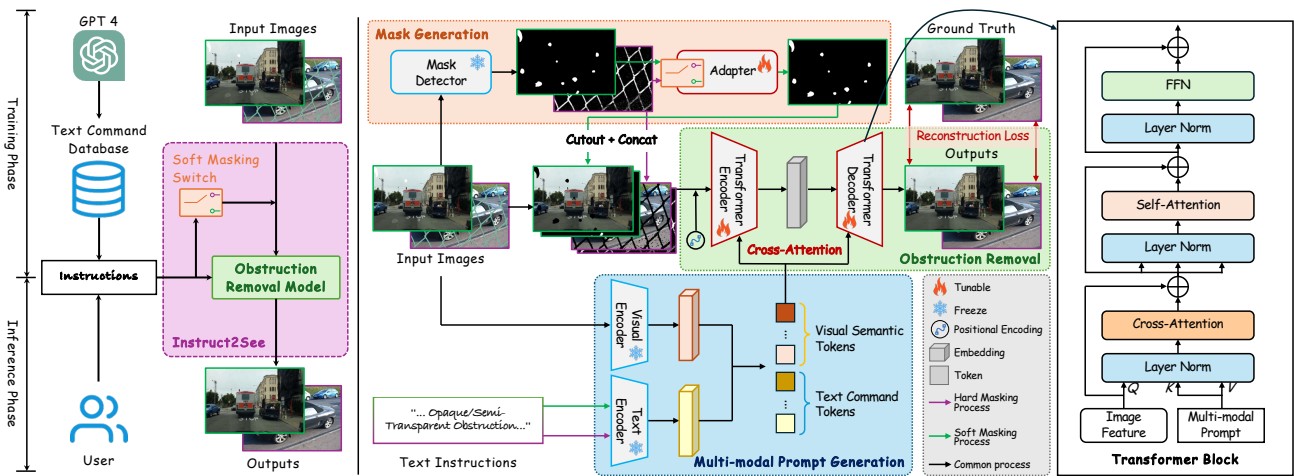

Figure 3: Flowchart of our Instruct2See. Instruct2See accepts instructions (randomly sampled from a database of instructions generated by GPT-4 during the training phase or inputted by the user during the inference phase) to flexibly activate soft masking and control the obstruction removal model for optimal capabilities.

tions, $B(x)$ is the underlying background image to be recovered, $R(x)$ denotes the obstruction components, and $M(x)$ is a binary mask. Here, $x$ refers to the pixel index. This formulation enables the decomposition of the input image into background and obstruction components using the mask $M(x)$ to separate them.

**Generalization to Unseen Obstructions.** As outlined in Eq. (1), obstruction removal is inherently an ill-posed problem, as it involves estimating the background scene $B(x)$ from a single input image $I(x)$ while accounting for the obstructions $R(x)$ represented by the mask $M(x)$. Most deep learning-based methods tackle this challenge by taking $I(x)$ as input and predicting $B(x)$ as output, relying on a network to learn the complex mapping from $I(x) \to B(x)$. However, these approaches are heavily data-dependent and typically perform well only on obstructions within the training distribution, becoming less effective when encountering out-of-distribution obstructions. This limitation arises primarily from the significant variation in the obstruction component $R(x)$ across different classes.

To address this challenge, our approach focuses on improving the model's ability to generalize by explicitly considering the variability in the obstruction component $R(x)$. By designing the model to handle a wide range of obstruction types beyond the training data, we enhance its capacity to perform well on unseen obstructions. This enables the model to maintain robust performance even when faced with occlusions that deviate significantly from those encountered during training.

## 4. Proposed Method

Fig. 3 outlines the flowchart of the proposed method. Unlike existing obstruction removal approaches that directly use $I$

in Eq. (1) as the model input, we first aim to mitigate the negative impact of $R$ on model generalization. We introduce $\hat{I}$ as the input to the restoration network, defined as:

$$\hat{I}(x) = I(x) - R(x) \circ M(x). \tag{2}$$

To achieve this, we utilize a trained mask detector $\mathcal{D}(\cdot)$ to estimate the mask $M$ from $I$. This estimated mask is used as the final mask for hard masking, while a tunable adapter $\mathcal{A}(\cdot)$ (see Sec. 4.1) is employed for soft masking, effectively compensating for inaccuracies in $M$ during the restoration process. The process is formulated as:

$$M(x) \approx \hat{M}(x) = \begin{cases} \mathcal{A}(\mathcal{D}(I(x))), & \text{if soft masking,} \\ \mathcal{D}(I(x)), & \text{if hard masking.} \end{cases} \tag{3}$$

Using $\hat{M}$, we remove the obstruction $R(x)$ from the original image $I$ to obtain $\hat{I}$. With this pre-processing, the obstruction removal task is simplified to:

$$\hat{I}(x) = B(x) \circ (1 - \hat{M}(x)). \tag{4}$$

To reconstruct the clear background scene, we develop a Transformer-based obstruction removal framework (detailed in Sec. 4.3) that learns the mapping $(\hat{I}, \hat{M}, T) \to B$ with $T$ being the text instruction. Multi-modal prompts (see Sec. 4.2) are integrated using a cross-attention unit to guide the reconstruction process. Finally, the network parameters $\Theta$ are optimized by minimizing the objective function $E(\cdot)$:

$$E(\Theta) = \frac{1}{N} \sum_{n=1}^{N} \left| B - f(\hat{I}, \hat{M}, T; \Theta) \right|, \tag{5}$$

where $N$ is the number of images, and $f(\cdot)$ represents the restoration operation.

### 4.1. Mask Generation

**Hard Masking and Motivation.** As shown by the purple process in Fig. 3, for obstructions with clear boundaries, such as fences, we apply an accurate mask to explicitly mark the regions requiring restoration, termed *hard masking*. However, this approach struggles when dealing with obstructions with ambiguous boundaries, such as raindrops, as it lacks flexibility in handling uncertain occlusion regions. This limitation ultimately affects restoration performance. To overcome this, we propose a *soft masking* approach, illustrated by the green process in Fig. 3, enabling the model to autonomously adjust the mask based on the obstruction's characteristics.

**Tunable Adapter for Soft Masking.** The tunable adapter is designed to improve reconstruction performance by mitigating the negative impact of inaccurate boundary estimates. Specifically, the tunable adapter takes the initial mask, estimated by the mask detector or manually provided, as input and outputs an optimized mask. The input first passes through a convolutional layer with batch normalization and ReLU activation. Subsequently, multiple Transformer blocks, incorporating self-attention units and feed-forward networks, are used to extract relevant features. A final convolutional layer produces the output mask. The key role of the adapter is to dynamically adjust the mask, allowing the model to determine the extent and regions where the mask should be applied based on the image features and occlusion conditions. This adaptive mechanism enhances flexibility, enabling selective restoration without strict reliance on the initial mask regions.

### 4.2. Multi-modal Prompt Generation

Using only $\hat{I}$ and $\hat{M}$ as inputs presents two key challenges: *1) the model lacks understanding of the required masking strategy for targeted restoration*, and *2) it struggles to extract high-level semantic information from the incomplete image, especially when encountering unseen obstructions*, leading to less accurate results. To address these issues, we introduce a multi-modal prompting strategy that leverages both text instructions and visual semantic embeddings to guide the image restoration process. Specifically, we input the text instruction $T$ and the original image $I$ into the CLIP model's text and visual encoders ($\Gamma_t$, $\Gamma_v$) to generate respective embeddings, which are concatenated to form the multi-modal prompt $P \in \mathbb{R}^L$, where $L$ is the number of tokens. This can be expressed as:

$$P = \text{concat}[\Gamma_t(T), \Gamma_v(I(x))] \in \mathbb{R}^L. \quad (6)$$

A cross-attention mechanism is then applied to integrate these prompts with the image features, guiding the reconstruction process. By incorporating text prompts, the model's ability to adapt its masking strategy improves, while visual prompts help prevent overfitting and enhance zero-shot generalization, reducing dependence on specific training data.

**Soft Masking Switch.** Based on the input instructions, we use the text embeddings $\Gamma_t(T)$ to activate the soft masking mode. Specifically, let the text representations for semi-transparent and opaque obstacles be denoted as $\Gamma_t(T_s)$ and $\Gamma_t(T_o)$, respectively; the similarity between these representations and the $\Gamma_t(T)$ can be expressed as follows:

$$\text{sim}(T, T_i) = \frac{\Gamma_t(T) \cdot \Gamma_t(T_i)}{\|\Gamma_t(T)\|\|\Gamma_t(T_i)\|}, \quad \text{for } i \in \{o, s\}. \quad (7)$$

After calculating the similarity index, we apply the Softmax function to normalize the vector such that the sum of its probabilities equals 1. When the similarity between $\Gamma_t(T)$ and $\Gamma_t(T_s)$ exceeds a threshold $\theta$, we enable the adapter strategy for adaptive adjustment of the mask.

### 4.3. Obstruction Removal

We develop a restoration network based on Restormer (Zamir et al., 2022), utilizing a Transformer-based encoder-decoder architecture for image restoration tasks.

**Cross-attention for Multi-modal Prompts.** To efficiently use the prompt information, we integrate cross-attention mechanisms into each Transformer block. By concatenating the embeddings from the text and visual prompts and feeding them into the Transformer blocks, we improve the model's ability to leverage multi-modal cues during the restoration process. The cross-attention unit is defined as:

$$\text{Cross-Att}(Q, K_p, V_p) = \text{Softmax}\left(\frac{Q \cdot K_p^\top}{\lambda}\right) V_p, \quad (8)$$

where $\lambda$ is a temperature factor, and $K_p$ and $V_p$ represent the key and value obtained from the multi-modal prompt, while $Q$ represents the query derived from the degraded image feature.

## 5. Experiments and Analysis

### 5.1. Experiment Settings

**Implementation Details.** Our Instruct2See framework is implemented in PyTorch 1.12.0 and trained on a system equipped with 2 AMD EPYC 7543 32-Core CPUs and 8 NVIDIA L40 GPUs. We train the model using the AdamW optimizer ($\beta_1 = 0.9$, $\beta_2 = 0.999$, weight decay of $1 \times 10^{-4}$) and L1 loss, over 300K iterations. The initial learning rate is set to $3 \times 10^{-4}$. A progressive learning strategy is employed, starting with a patch size of $128 \times 128$ and a batch size of 1. The patch size is progressively updated to $128 \times 128$,

Table 1: PSNR, SSIM, CLIP Score (CLIPS), and LPIPS comparisons of different methods on *seen* obstructions. The best and second best results are highlighted in **bold** and underlined.

| Method | Venue | Fence | | | | Flare | | | | Raindrop | | | | Average | | | |
|---|---|---|---|---|---|---|---|---|---|---|---|---|---|---|---|---|---|
| | | PSNR↑ | SSIM↑ | CLIPS↑ | LPIPS↓ | PSNR↑ | SSIM↑ | CLIPS↑ | LPIPS↓ | PSNR↑ | SSIM↑ | CLIPS↑ | LPIPS↓ | PSNR↑ | SSIM↑ | CLIPS↑ | LPIPS↓ |
| Restormer | CVPR22 | 29.86 | **0.9170** | 0.9045 | 0.0986 | 25.41 | 0.9162 | 0.9547 | 0.0772 | 30.07 | 0.9542 | 0.9670 | 0.0482 | 28.45 | 0.9291 | 0.9421 | 0.0746 |
| TransWeather | CVPR22 | 26.93 | 0.8492 | 0.8964 | 0.1157 | 25.18 | 0.9040 | 0.9323 | 0.0824 | 30.44 | 0.9508 | 0.9468 | 0.0787 | 27.52 | 0.9013 | 0.9252 | 0.0922 |
| PromptIR | NeurIPS23 | 24.59 | 0.7423 | 0.8957 | 0.1521 | 25.43 | 0.9187 | **0.9604** | 0.0721 | 31.95 | 0.9668 | 0.9679 | 0.0471 | 27.32 | 0.8759 | 0.9413 | 0.0904 |
| WGWSNet | CVPR23 | 23.19 | 0.7878 | 0.8561 | 0.2884 | **25.87** | 0.9192 | 0.9486 | 0.0710 | **32.89** | 0.9671 | 0.9706 | 0.0486 | 27.32 | 0.8914 | 0.9251 | 0.1360 |
| Histoformer | ECCV24 | 28.05 | 0.9001 | 0.8971 | 0.1134 | 25.19 | 0.9195 | 0.9541 | 0.0712 | 31.59 | 0.9614 | 0.9584 | 0.0548 | 28.28 | 0.9270 | 0.9365 | 0.0798 |
| XRestormer | ECCV24 | 27.11 | 0.8972 | 0.9037 | 0.1430 | 24.89 | 0.9185 | 0.9487 | 0.0743 | 30.55 | 0.9583 | 0.9557 | 0.0482 | 27.52 | 0.9247 | 0.9360 | 0.0924 |
| Instruct2See | ICML25 | **30.15** | 0.9079 | **0.9093** | **0.0941** | 25.15 | **0.9202** | 0.9509 | **0.0694** | 32.64 | **0.9680** | **0.9723** | **0.0446** | **29.31** | **0.9320** | **0.9442** | **0.0694** |

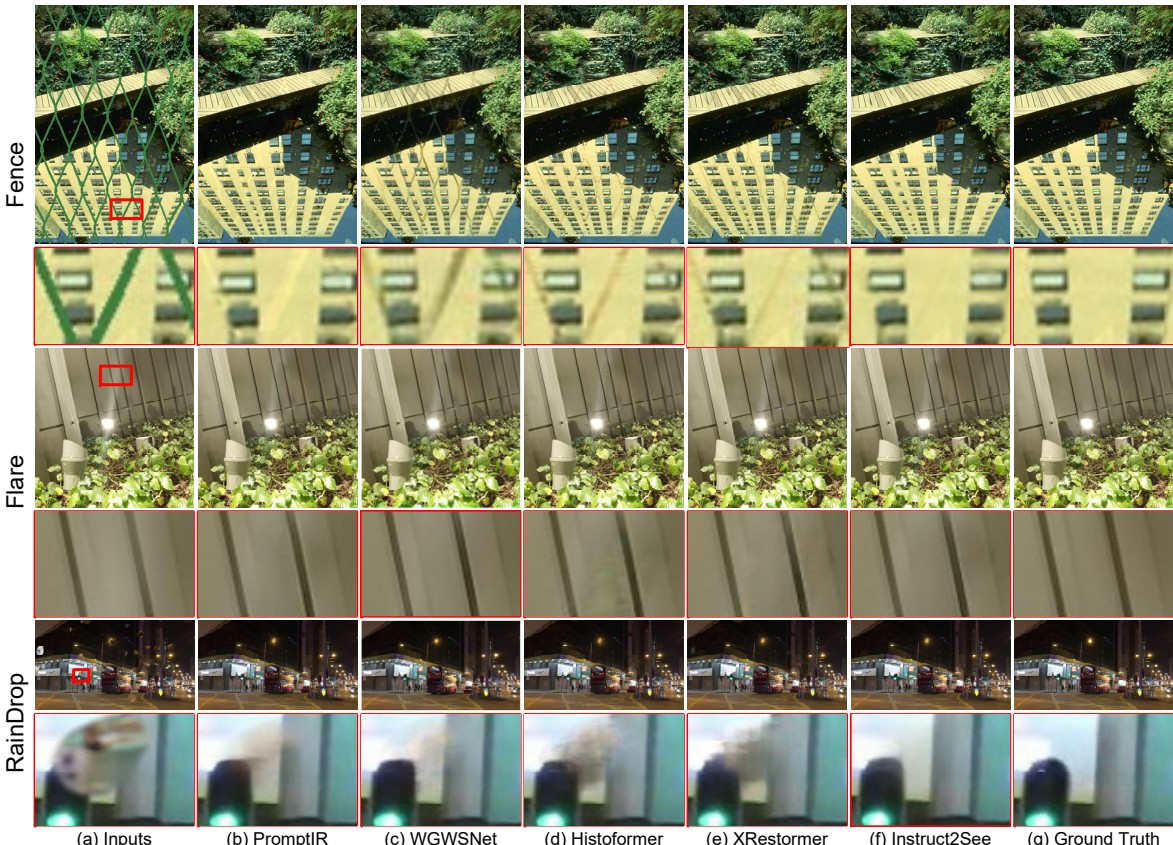

(a) Inputs (b) PromptIR (c) WGWSNet (d) Histoformer (e) XRestormer (f) Instruct2See (g) Ground Truth

Figure 4: Visual comparisons of our method with other approaches on *seen* obstructions.

$160 \times 160$, $192 \times 192$, and $256 \times 256$ at iterations 115,000, 80,000, 60,000, and 45,000, respectively. We also apply horizontal and vertical flips for data augmentation.

**Datasets.** We utilize 3,984 images for model training. For the fence obstacle, we select 897 clear images from the BSD dataset (Martin et al., 2001) and the UCID dataset (Schaefer & Stich, 2003), and generate paired data using the fence synthesis method from (Du et al., 2018). Additionally, 987 clear images from the Flickr24K dataset (Zhang et al., 2018) and 987 flare images from the Flare7K dataset (Dai et al., 2022) are used to create flare image pairs. We also include 2,100 training image pairs from the VRDS dataset (Wu et al., 2023). For testing, we apply the same synthesis strategy to create a fence test dataset with 100 image pairs. Moreover, a flare test dataset with another 100 image pairs is used.

Additionally, 500 raindrop test image pairs are included. For unseen obstructions, we sourced 100 test images each from the rain streak dataset (Yang et al., 2017), snowy dataset (Liu et al., 2018), and stroke dataset (Lugmayr et al., 2022). We also tested our method on special obstacles to verify its zero-shot capability.

### 5.2. Comparisons with the State-of-the-Arts

**Results on Seen Obstructions.** The quantitative evaluation results for seen obstructions are presented in Table 1. We compare the obstruction removal performance of various methods by using detected masks. While our proposed method is slightly outperformed by certain state-of-the-art methods in specific tasks based on PSNR, SSIM,

Table 2: PSNR, SSIM, CLIP Score (CLIPS), and LPIPS comparisons of different methods on *unseen* obstructions. The best and second best results are highlighted in **bold** and underlined.

| Method | Venue | Rain Streak | | | | Snow | | | | Stroke | | | | Average | | | |
|---|---|---|---|---|---|---|---|---|---|---|---|---|---|---|---|---|---|
| | | PSNR↑ | SSIM↑ | CLIPS↑ | LPIPS↓ | PSNR↑ | SSIM↑ | CLIPS↑ | LPIPS↓ | PSNR↑ | SSIM↑ | CLIPS↑ | LPIPS↓ | PSNR↑ | SSIM↑ | CLIPS↑ | LPIPS↓ |
| Restormer | CVPR22 | 26.19 | 0.8381 | 0.9133 | 0.2385 | 28.81 | 0.9013 | 0.9355 | 0.1508 | 21.14 | 0.8173 | 0.8012 | 0.2061 | 25.38 | 0.8522 | 0.8833 | 0.1984 |
| TransWeather | CVPR22 | 26.70 | 0.8341 | 0.8989 | 0.2450 | 29.53 | 0.8926 | 0.9455 | 0.1317 | 18.27 | 0.6752 | 0.8014 | 0.3598 | 24.83 | 0.8006 | 0.8819 | 0.2455 |
| PromptIR | NeurIPS23 | 24.04 | 0.7197 | 0.8869 | 0.2211 | 26.01 | 0.7457 | 0.9370 | 0.1185 | 29.39 | 0.9021 | 0.8533 | 0.0970 | 26.48 | 0.7892 | 0.8924 | 0.1455 |
| WGWSNet | CVPR23 | 29.68 | **0.9111** | **0.9422** | **0.1411** | 29.54 | 0.8944 | 0.9439 | 0.1273 | 28.25 | 0.8722 | **0.8791** | 0.1222 | 29.18 | 0.8927 | **0.9217** | 0.1302 |
| Histoformer | ECCV24 | 27.99 | 0.8634 | 0.8854 | 0.1870 | 32.40 | 0.9203 | 0.9421 | 0.0909 | 28.07 | 0.8761 | 0.8379 | 0.1206 | 29.49 | 0.8866 | 0.8885 | 0.1328 |
| XResrormer | ECCV24 | 28.05 | 0.8560 | 0.9138 | 0.2108 | 31.31 | 0.9170 | 0.9567 | 0.1040 | 19.00 | 0.7588 | 0.7754 | 0.2472 | 26.12 | 0.8439 | 0.8820 | 0.1873 |
| Instruct2See | ICML25 | **29.82** | 0.8907 | 0.9296 | 0.1639 | **34.85** | **0.9283** | **0.9618** | **0.0639** | **29.45** | **0.9067** | 0.8507 | **0.0936** | **31.37** | **0.9086** | 0.9140 | **0.1071** |

CLIPS, and LPIPS metrics, the overall results clearly highlight the strengths and advantages of our approach. Notably, our method achieves a PSNR that is 0.86 dB higher than the second-best method, Restormer, demonstrating its superior ability to preserve image quality. Furthermore, as shown in Fig. 4, visual comparisons across the three obstruction removal tasks consistently emphasize the strengths of our approach, particularly in reconstructing fine details and maintaining scene coherence. For additional numerical experiments under ideal conditions (using ground truth masks), please refer to the Appendix.

**Results on Unseen Obstructions.** To further evaluate the zero-shot learning capability of our model, we conducted experiments on images containing unseen obstructions. Table 2 presents the PSNR, SSIM, CLIPS, and LPIPS results for obstruction removal on three classic obstacles not included in the training. With the exception of a slightly lower CLIPS compared to WGWSNet, our method consistently delivers the best results across the unseen obstruction. As shown in Fig. 5, visual comparisons across additional obstructions, such as yarn, scratches, spots, and power cables, further demonstrate the strong generalization ability of our proposed method. While existing methods often struggle or show limited effectiveness in addressing out-of-distribution obstructions, our approach consistently produces more accurate and realistic restorations. This confirms the performance boost provided by our multi-modal prompt strategy and tunable adapter, which enable effective zero-shot learning and allow our model to capture the nuances of unseen obstructions. These results validate the robustness and flexibility of our method, making it a promising solution for real-world applications where diverse and unpredictable obstructions are common.

In addition, the inpainting- and editing-based methods exhibit strong zero-shot capabilities and perform well in various object removal or image editing tasks, making them viable solutions for the obstruction removal task. Therefore, for a comprehensive comparison, we evaluated our proposed method against two inpainting methods (LaMa (Suvorov et al., 2022) and RePaint (Lugmayr et al., 2022)) and one editing method (DiffEdit (Couairon et al., 2023))

Table 3: PSNR and SSIM comparisons of our method with inpainting/editing-based methods on *unseen* obstructions. The best results are highlighted in **bold**.

| Method | Venue | Rain Streak | | Snow | | Stroke | | Average | |
|---|---|---|---|---|---|---|---|---|---|
| | | PSNR↑ | SSIM↑ | PSNR↑ | SSIM↑ | PSNR↑ | SSIM↑ | PSNR↑ | SSIM↑ |
| LaMa | WACV22 | 29.07 | 0.8858 | 32.32 | 0.9108 | 28.10 | 0.8728 | 29.83 | 0.8898 |
| RePaint | CVPR22 | 28.78 | 0.8865 | 32.20 | 0.9064 | 23.78 | 0.8059 | 28.25 | 0.8662 |
| DiffEdit | ICLR23 | 23.88 | 0.6561 | 24.23 | 0.6732 | 11.65 | 0.6072 | 19.92 | 0.6455 |
| Instruct2See | ICML25 | **29.82** | **0.8907** | **34.85** | **0.9283** | **29.45** | **0.9067** | **31.37** | **0.9086** |

in the zero-shot obstruction removal. Table 3 presents the quantitative evaluation results for these methods and ours across three classic obstacle removal scenarios. In particular, the editing-based method (DiffEdit) has poor numerical performance, while inpainting-based methods fail to accurately reconstruct details. These stem from their design as image inpainting and editing tools, which strictly constrain changes to masked regions and lack the flexibility to handle irregular-shaped holes. The results indicate that, despite using obstacle masks as inputs, existing inpainting and editing methods still struggle to effectively address this problem. In contrast, our method, which incorporates a tunable mask adapter and multimodal feature representation of obstacles, demonstrates superior performance in zero-shot obstacle removal tasks. Fig. 6 further visualizes additional obstacle removal results. It is evident that while LaMa and RePaint exhibit some obstacle removal capabilities, residual obstacles remain. Conversely, our Instruct2See effectively handles various situations and robustly removes obstacles.

### 5.3. Ablation Study

**Effectiveness of Network Modules.** Table 4 (a) presents a comprehensive quantitative evaluation of different module configurations on three seen and three unseen obstructions. The results clearly show that the baseline model alone produces poor results, as indicated by the low PSNR and SSIM scores. Introducing a mask improves performance slightly, but the enhancement remains limited. This is primarily because the model struggles to distinguish between different types of obstructions and cannot effectively select the appropriate masking strategy. Additionally, without a proper

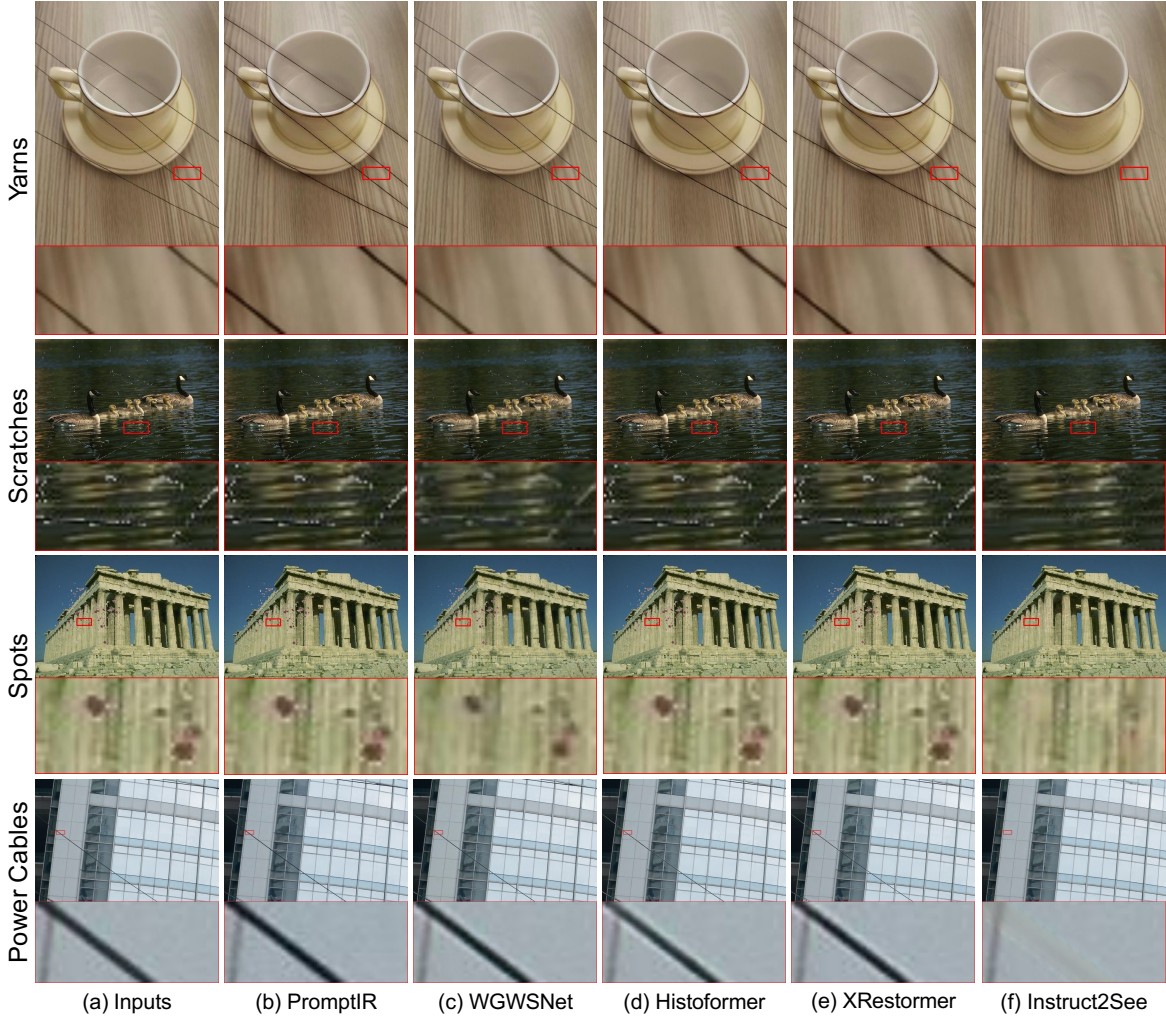

Figure 5: Visual comparisons of our method with other approaches on *unseen* obstructions.

understanding of scene semantics, the model generates unrealistic and anomalous restoration outcomes. In contrast, incorporating the cross-attention mechanism, which integrates textual instructions with visual semantics, significantly improves performance, leading to a PSNR increase of 1.95 and an SSIM increase of 0.0113. This mechanism enables the model to better grasp the contextual relationship between the obstruction and the surrounding scene, producing more coherent and realistic restoration results. Finally, the tunable adapter can further enhance the ability to handle obstructions with blurred boundaries, resulting in optimal effects across all metrics. This verified that our model can adaptively manage different obstruction scenarios, leading to a more refined and effective restoration process.

**Effectiveness of Different Prompts.** Table 4 (b) compares performance using different prompt strategies. Without prior prompts, the model performs poorly, primarily due to confusion over the correct masking strategy and a lack of semantic understanding, especially for unseen obstructions.

Table 4: PSNR and SSIM comparisons of integrating different modules (a) and using different prompt strategies (b). $P_t$ and $P_v$ represent text and visual prompts, respectively.

| mask | CA | Adapter | PSNR↑ | SSIM↑ |
|------|----|---------|-------|-------|
|      |    |         | 27.05 | 0.8920 |
| ✓    |    |         | 28.05 | 0.9004 |
| ✓    | ✓  |         | 30.00 | 0.9117 |
| ✓    | ✓  | ✓       | 30.93 | 0.9250 |

(a)

| $P_t$ | $P_v$ | PSNR↑ | SSIM↑ |
|-------|-------|-------|-------|
|       |       | 28.65 | 0.9063 |
| ✓     |       | 29.73 | 0.9168 |
|       | ✓     | 30.25 | 0.9215 |
| ✓     | ✓     | 30.93 | 0.9250 |

(b)

When using only the textual embedding, the model can adopt the correct strategy to handle both sharp and blurred mask boundaries. However, due to the absence of complete image semantics from occluded regions, the model often produces unrealistic or inconsistent results. Using only the visual encoder strategy can better compensate for the semantic loss caused by obstacle removal, thereby achieving better results than introducing only the text encoder. Finally, by integrating both visual semantics and textual instructions through a multi-modal prompt, the model can easily handle obstruction removal tasks for both in-distribution and out-of-

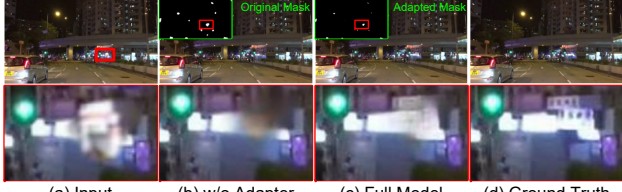

(a) Input     (b) w/o Adapter     (c) Full Model     (d) Ground Truth

Figure 8: Visual comparisons of using our tunable adapter on a raindrop obstruction case.

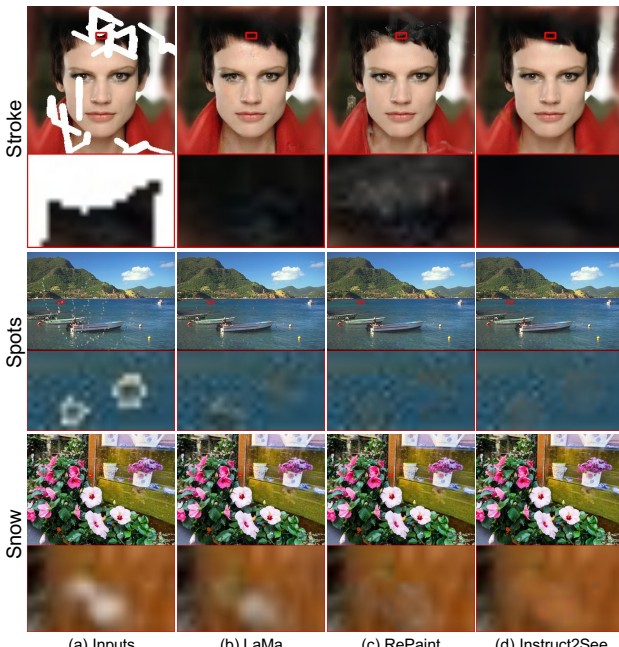

(a) Inputs     (b) LaMa     (c) RePaint     (d) Instruct2See

Figure 6: Visual comparisons of our method with inpainting methods on *unseen* obstructions.

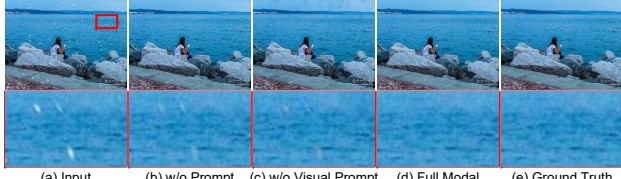

(a) Input   (b) w/o Prompt   (c) w/o Visual Prompt   (d) Full Modal   (e) Ground Truth

Figure 7: Visual comparisons of using different prompt strategies on a snow obstruction case.

distribution obstacles. The multi-modal prompt strategy not only improves the model's interpretability but also strengthens its ability to generalize to unfamiliar obstructions.

To further illustrate the impact of multi-modal prompts, we compared different strategies in a snow obstruction case, as shown in Fig. 7. Without any prompt, the model shows only a slight reduction of the snow obstacles. Introducing a textual prompt (Fig. 7(c)) allows the model to focus more on masking strategy, leading to better suppression of the obstruction. However, using only the textual prompt introduces unnatural artifacts in the reconstruction, as the model lacks complete semantic information from the occluded regions. By incorporating the visual encoder from the visual-language model, we effectively compensate for the missing semantic details during obstruction removal. This approach preserves the model's robust zero-shot learning capability while enabling it to extract relevant details and accurately represent various obstructions, even those not encountered during training. Consequently, the multi-modal prompt strategy delivers superior visual restoration and enhanced obstruction suppression performance.

**Effectiveness of Tunable Adapter.** We designed a tunable

adapter for soft masking to address inaccuracies in occlusion regions. This adapter dynamically adjusts the mask, enabling our model to determine the extent of mask application based on the image features, rather than being confined to a predefined mask area. To evaluate the function and effectiveness of our proposed adapter, we conducted an ablation study. As shown in Fig. 8, a comparison between the adjusted and original masks demonstrates that the adapter effectively refines the mask for uncertain obstructions, such as raindrops. The original mask often overly covers the restoration area, leading to a loss of detail and suboptimal reconstruction. In contrast, the adjustments made by the tunable adapter ensure more accurate and reliable restoration by preserving crucial details in the occluded regions. Additionally, the ablation study reveals that the tunable adapter significantly improves the model's adaptability to various obstructions with ambiguous boundaries, resulting in a PSNR increase of 0.93 and an SSIM gain of 0.0133 compared to the fixed mask approach. These findings confirm that the tunable adapter not only optimizes mask coverage but also plays a vital role in refining the restoration process.

## 6. Conclusion

In this work, we proposed Instruct2See, a novel zero-shot obstruction removal framework designed to effectively address challenges posed by both in-distribution and out-of-distribution obstructions. By leveraging multi-modal prompts that integrate visual semantics and textual descriptions through a cross-attention mechanism, Instruct2See demonstrated superior performance in accurately reconstructing occluded scenes. The inclusion of a tunable adapter for soft masking further improved adaptability, allowing the model to handle ambiguous boundaries with greater flexibility. Extensive experiments validated the efficacy and generalization capabilities of Instruct2See, highlighting its potential as a robust solution for real-world obstruction removal tasks.

**Limitations.** Our method targets small, numerous obstacles (e.g., raindrops and fences) and explores optimal strategies for removing both opaque and translucent obstacles. It is not intended for large obstructions, as it focuses on scene recovery through contextual cues. In such cases, inpainting techniques may be more effective.

## Acknowledgements

This work is supported by the Guangdong Natural Science Funds for Distinguished Young Scholars (Grant 2023B1515020097), the GuangDong Basic and Applied Basic Research Foundation (2025A1515010124), the National Research Foundation, Singapore under its AI Singapore Programme (AISG Award No: AISG3-GV-2023-011), and the Lee Kong Chian Fellowships.

## Impact Statement

This paper presents work whose goal is to advance the field of Machine Learning. There are many potential societal consequences of our work, none of which we feel must be specifically highlighted here.

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

# A. Implementation Details

## A.1. Model Inference

As outlined in Algorithm 1, our model operates in two modes during inference: opaque obstruction removal using hard masking and semi-transparent obstruction removal using soft masking. The process begins with the image $I$, containing the obstruction, and a text instruction $T$. The overall obstruction removal procedure is divided into three key steps: mask generation, multi-modal prompt generation, and obstruction removal.

---

**Algorithm 1** Instruct2See Model Inference

---

1:   $I$: input image, $B$ clear background, $\bar{M}$: initial mask, $\hat{M}$: adapted mask, $\hat{I}$: input image cutout by $\hat{M}$, $T$: text instruction, $T_s$: "semi-transparent obstacle" text, $\mathcal{D}(\cdot)$: mask detector, $\mathcal{A}(\cdot)$: tunable adapter, $\mathcal{R}(\cdot, \cdot, \cdot)$: Obstruction Eliminator, $\Gamma_t(\cdot)$: visual language model's text encoder, $\Gamma_v(\cdot)$: visual language model's visual encoder, $P_t$: text prompt, $P_v$: visual prompt, $P$: multi-modal prompt, $concat$: embedding splicing operation

**Input:** $I, T$

2:   $\bar{M} \leftarrow \mathcal{D}(I)$ {Initial mask generation.}
3:   $P_t \leftarrow \Gamma_t(T)$
4:   $P_v \leftarrow \Gamma_v(I)$
5:   **if** $\text{sim}(P_t, \Gamma_t(T_s)) > \theta$ **then**
6:      $\hat{M} \leftarrow \mathcal{A}(\bar{M})$ {Soft masking.}
7:   **else**
8:      $\hat{M} = \bar{M}$ {Hard masking.}
9:   **end if**
10:   $P = concat[P_t, P_v]$ {Multi-modal prompt generation.}
11:   get $\hat{I}$ by cutting out the region in $\hat{M}$ from $I$
12:   $B \leftarrow \mathcal{R}(\hat{I}, \hat{M}, P)$ {Obstruction removal.}
13:   **return** $B$

---

**Mask Generation.** We first extract the initial mask $\bar{M}$ from the input image $I$ using a mask detector (as described in line 2 of Algorithm 1). For obstructions like rain streaks and snow, which are more challenging to segment, we employ a U-Net-based model (Ronneberger et al., 2015) to generate the initial mask. For other obstructions, we use the Segment Anything Model 2 (SAM2) (Ravi et al., 2024). Depending on the type of obstruction, different masking strategies are applied: for opaque obstructions with clear boundaries, we directly use $\bar{M}$ as the final mask $\hat{M}$ (lines 7–9), while for semi-transparent obstructions with blurred edges, we refine $\bar{M}$ using a tunable adapter to improve performance (lines 5–7).

**Multi-Modal Prompt Generation.** We process the inputs $I$ and $T$ using the text and image encoders of the Contrastive Language-Image Pre-training (CLIP) model (Radford et al., 2021) to obtain textual and visual embeddings $(P_t, P_v)$. These embeddings are then concatenated to generate the multi-modal prompt $P$, as described in lines 3, 4, and 10 of Algorithm 1.

**Obstruction Removal.** With the refined mask $\hat{M}$ and the multi-modal prompt $P$, we first use $\hat{M}$ to mask out the obstructions in $I$, generating $\hat{I}$. Then, $\hat{I}$ and $\hat{M}$ are concatenated along the channel dimension and, along with $P$, input into the obstruction removal model $\mathcal{R}(\cdot)$ (lines 11–13). A cross-attention module within $\mathcal{R}(\cdot)$ fuses the image features with the multi-modal prompt. Specifically, features from the image map are extracted using convolution as the query, while the multi-modal prompt generates key and value vectors via two independent linear layers. These vectors are fused using a multi-head attention mechanism, guiding the network to effectively remove unknown obstructions.

## A.2. CLIP Usage

In the use of CLIP[1], the visual encoder is employed to extract visual features from the original image to compensate for the loss of visual semantics caused by obstacle cutout. Since the pre-trained CLIP visual encoder already possesses strong semantic representation capabilities, we do not perform additional fine-tuning on this module. The text embeddings, however, provide specific removal prompts to the model. Due to the relatively few specific instructions for obstacle removal in CLIP's original training, the original embedding space may not be suitable for this task (i.e., the embeddings generated by text instructions for the same goal may exhibit significant variability). Therefore, we only fine-tune CLIP's text encoder. The

---

[1]We utilize the CLIP ViT-B/32 model.

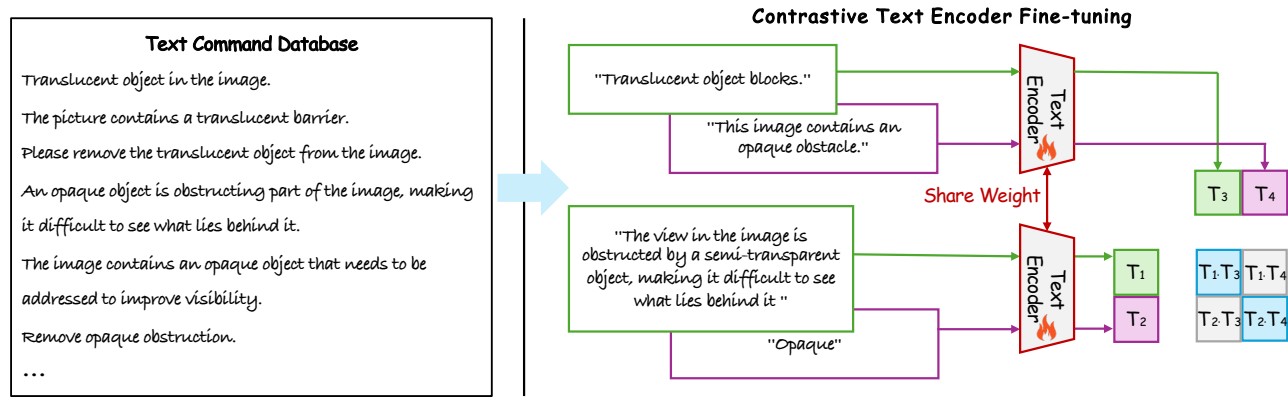

Figure 9: Contrastive fine-tuning of CLIP text encoder.

Table 5: PSNR and SSIM comparisons of different methods on *seen* obstructions. The scheme using the GT mask as input is designed to demonstrate the obstruction removal capabilities of each model under ideal conditions.

| Method | Fence | | Flare | | Raindrop | | Average | |
|---|---|---|---|---|---|---|---|---|
| | PSNR↑ | SSIM↑ | PSNR↑ | SSIM↑ | PSNR↑ | SSIM↑ | PSNR↑ | SSIM↑ |
| Restormer | 29.86 | 0.9170 | 25.41 | 0.9162 | 30.07 | 0.9542 | 28.45 | 0.9291 |
| *w GT mask* | 29.62 | 0.9166 | 25.38 | 0.9145 | 32.04 | 0.9651 | 29.01 | 0.9321 |
| TransWeather | 26.93 | 0.8492 | 25.18 | 0.9040 | 30.44 | 0.9508 | 27.52 | 0.9013 |
| *w GT mask* | 29.12 | 0.8727 | 26.05 | 0.9150 | 30.58 | 0.9510 | 28.58 | 0.9129 |
| PromptIR | 24.59 | 0.7423 | 25.43 | 0.9187 | 31.95 | 0.9668 | 27.32 | 0.8759 |
| *w GT mask* | 26.41 | 0.7842 | 25.63 | 0.9193 | 32.71 | 0.9691 | 28.25 | 0.8909 |
| WGWSNet | 23.19 | 0.7878 | 25.87 | 0.9192 | 32.89 | 0.9671 | 27.32 | 0.8914 |
| *w GT mask* | 26.88 | 0.8467 | 25.50 | 0.9193 | 32.26 | 0.9648 | 28.21 | 0.9103 |
| Histoformer | 28.05 | 0.9001 | 25.19 | 0.9195 | 31.59 | 0.9614 | 28.28 | 0.9270 |
| *w GT mask* | 32.29 | 0.9382 | 25.96 | 0.9106 | 32.29 | 0.9636 | 30.18 | 0.9375 |
| XRestormer | 27.11 | 0.8972 | 24.89 | 0.9185 | 30.55 | 0.9583 | 27.52 | 0.9247 |
| *w GT mask* | 30.57 | 0.8972 | 25.44 | 0.9176 | 31.02 | 0.9599 | 29.01 | 0.9249 |
| Instruct2See | 30.15 | 0.9079 | 25.15 | 0.9202 | 32.64 | 0.9680 | 29.31 | 0.9320 |
| *w GT mask* | 32.12 | 0.9329 | 25.83 | 0.9203 | 33.52 | 0.9706 | 30.49 | 0.9413 |

fine-tuning strategy for the CLIP text encoder is shown in Fig. 9. We first collected the text instructions corresponding to each image in our training dataset to construct a text command database. This database contains two categories: instructions for removing opaque obstacles and instructions for removing semi-transparent obstacles. Subsequently, we fine-tuned the model using a contrastive pre-training strategy similar to CLIP.

More specifically, in each iteration, we randomly select text instructions in the database and use the CLIP text encoder to generate two text embeddings for opaque obstacles ($T_2$ and $T_4$) and two text embeddings for semi-transparent obstacles ($T_1$ and $T_3$). Subsequently, we calculate the cosine similarity between each pair and designate the values calculated between the same category as positive samples, while the values calculated between different categories are designated as negative samples. Finally, we perform contrastive training based on the clip loss (Radford et al., 2021).

Additionally, the tunable adapter is only activated for semi-transparent obstacles. To selectively enable this function based on the input instruction, we set up two-word embeddings: "opaque" and "semi-transparent". By calculating the cosine similarity between the instructions embedding and these two embeddings, we can determine whether to activate the adapter module. Therefore, this fine-tuning strategy allows our model to more accurately judge when to enable the adapter.

## B. More Experimental Results

### B.1. More Numerical Results

To demonstrate the upper bound of seen obstacle removal under ideal conditions, we compared the restoration performance of various methods using both detected masks and ground-truth masks. As shown in Table 5, the results show that higher-quality masks benefit most methods. In our task, the mask detector is plug-and-play, and future adoption of more powerful

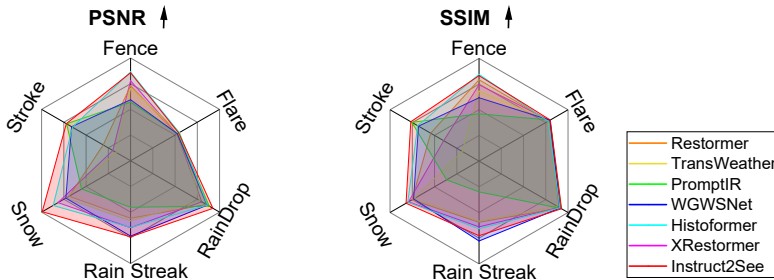

Figure 10: PSNR and SSIM comparisons of different methods on seen and unseen obstructions.

mask detectors could further enhance the obstruction removal capability. Additionally, we present a radar chart (Fig. 10) comparing the PSNR and SSIM results for three types of seen obstacles and three classic unseen obstacles. The results demonstrate that the proposed model delivers robust performance across various scenarios, validating its effectiveness and strong generalization ability.

## B.2. Zero-Shot Single Obstruction Removal

This section presents additional examples of removing various unseen obstructions. Fig. 11 compares the results of different methods on three typical obstruction removal tasks. It is evident that TransWeather and XRestormer perform poorly in the stroke removal task, failing to handle such cases effectively. Other methods also produce distorted facial details during restoration. For semi-transparent obstructions, such as raindrops and snow, these methods fail to properly capture the relationship between the obstruction and the mask, leading to ineffective or minimal removal.

In contrast, our method employs a hard-soft masking strategy, allowing smooth transitions between hard and soft masking. This enables it to capture the complexity and diversity of real-world occlusion scenarios more effectively. Fig. 12 showcases further experiments on uncommon obstructions, demonstrating the zero-shot generalization capability of our method. This advantage stems from our distribution-agnostic approach, which formulates obstruction removal as a soft-hard masking problem, representing any obstruction through the integration of visual semantic embeddings and textual instructions.

## B.3. Zero-Shot Multiple Obstruction Removal

Fig. 13 displays three visualization cases on multiple obstruction removal. It is evident that our method can accurately represent specified obstructions through multi-modal prompts and masks, and easily eliminate them using the designed model. From the results, it appears that only when there are occlusions between multiple obstacles does the elimination of one obstacle inevitably affect another. The order of obstruction elimination does not have a significant impact on the results.

Table 6: PSNR and SSIM comparisons of using different prompt generation strategies.

| Model | PSNR↑ | SSIM↑ |
|---|---|---|
| Instruct2See + CLIP | 30.93 | 0.9250 |
| Instruct2See + BLIP | 31.01 | 0.9235 |

## B.4. Metric Influence of Using Different Prompt Generation Models

In this section, we compared the effects of using CLIP's and BLIP's encoders to generate textual and visual embeddings on obstacle removal results. The experimental results are shown in Table 6. Clearly, whether using CLIP (Radford et al., 2021) or BLIP (Li et al., 2022b), our model can generate robust obstacle removal effects, proving that our model can adapt to commonly used pretrained encoders.

## C. Failure Cases using Incorrect Description

Fig. 14 illustrates two examples of using incorrect descriptions. In the stroke removal case, when an instruction of a semi-transparent obstruction removal is used for a scene with an opaque obstruction, our model tends to treat the original

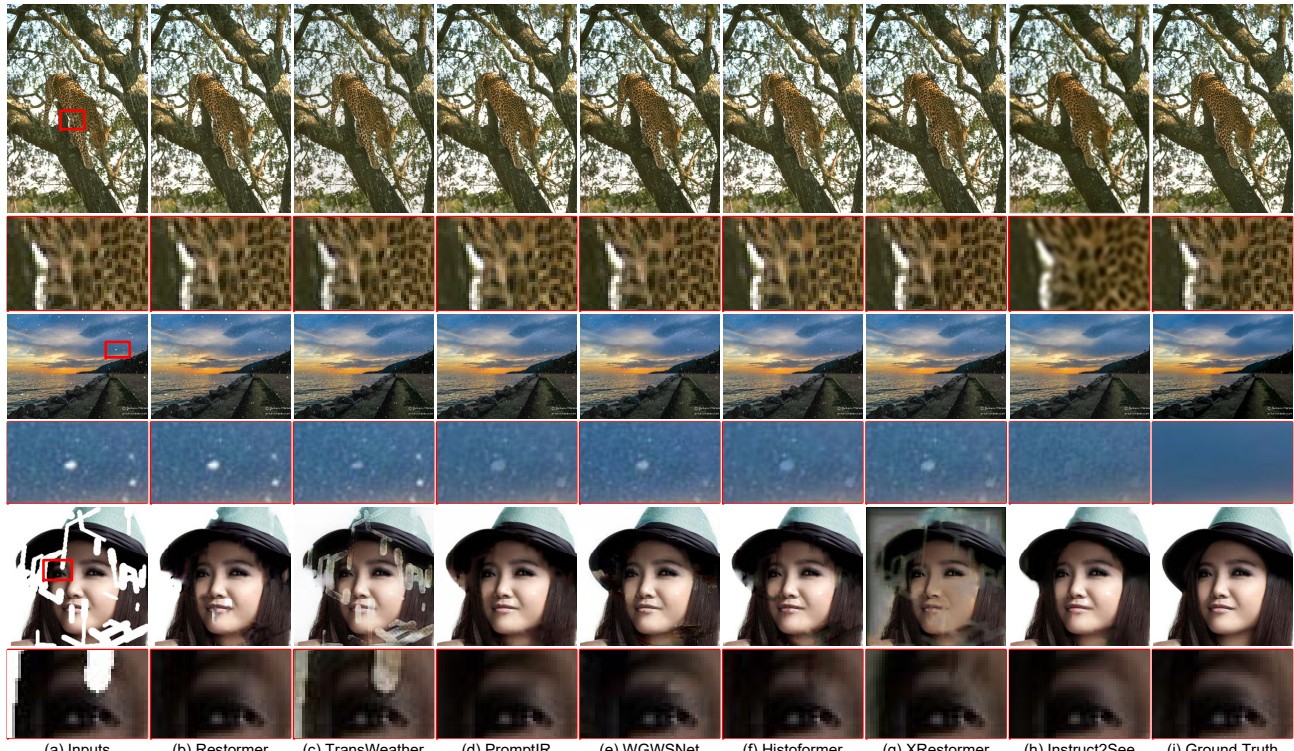

| (a) Inputs | (b) Restormer | (c) TransWeather | (d) PromptIR | (e) WGWSNet | (f) Histoformer | (g) XRestormer | (h) Instruct2See | (i) Ground Truth |

Figure 11: Visual comparisons on three classic unseen obstructions (rain streak, snow, and stroke).

opaque obstruction as part of the real information, leading to suboptimal results. Accurately describing the obstacle as an opaque obstacle can easily resolve this issue. Similarly, in the shadow removal case, using an opaque obstruction description will make the masked image content completely invisible, resulting in a restoration that does not match reality, especially in cases of large-area occlusions. However, correcting the instruction to remove the transparent obstacle can solve this problem.

## D. Complexity Analysis

In this section, we present the model sizes of various methods and calculate their Floating Point Operations (FLOPs) and runtime on 224x224 images. As shown in Table 7, although our model has the highest number of parameters due to the introduction of a cross-attention module integrated with multi-modal prompts and an adjustable mask adapter, its FLOPs and inference speed remain at a moderate level compared to competing methods. In the future, we will consider maintaining the model's strong zero-shot generalization capabilities while reducing computational costs.

Table 7: Comparisons of parameters, FLOPs, and runtime between.

| Model | Venue | Parameters (M) | FLOPs (G) | Runtime (ms) |
|---|---|---|---|---|
| Restormer | CVPR22 | 26.13 | 118.60 | 49.37±0.46 |
| TransWeather | CVPR22 | 38.06 | 3.57 | 19.64±0.05 |
| PromptIR | NeurIPS23 | 35.59 | 132.26 | 53.95±0.47 |
| WGWSNet | CVPR23 | 4.70 | 96.65 | 88.39±0.35 |
| Histoformer | ECCV24 | 16.62 | 86.79 | 83.13±0.82 |
| XRestromer | ECCV24 | 22.34 | 155.49 | 100.67±0.44 |
| Instruct2See | ICML25 | 56.69 | 146.23 | 84.28±0.61 |

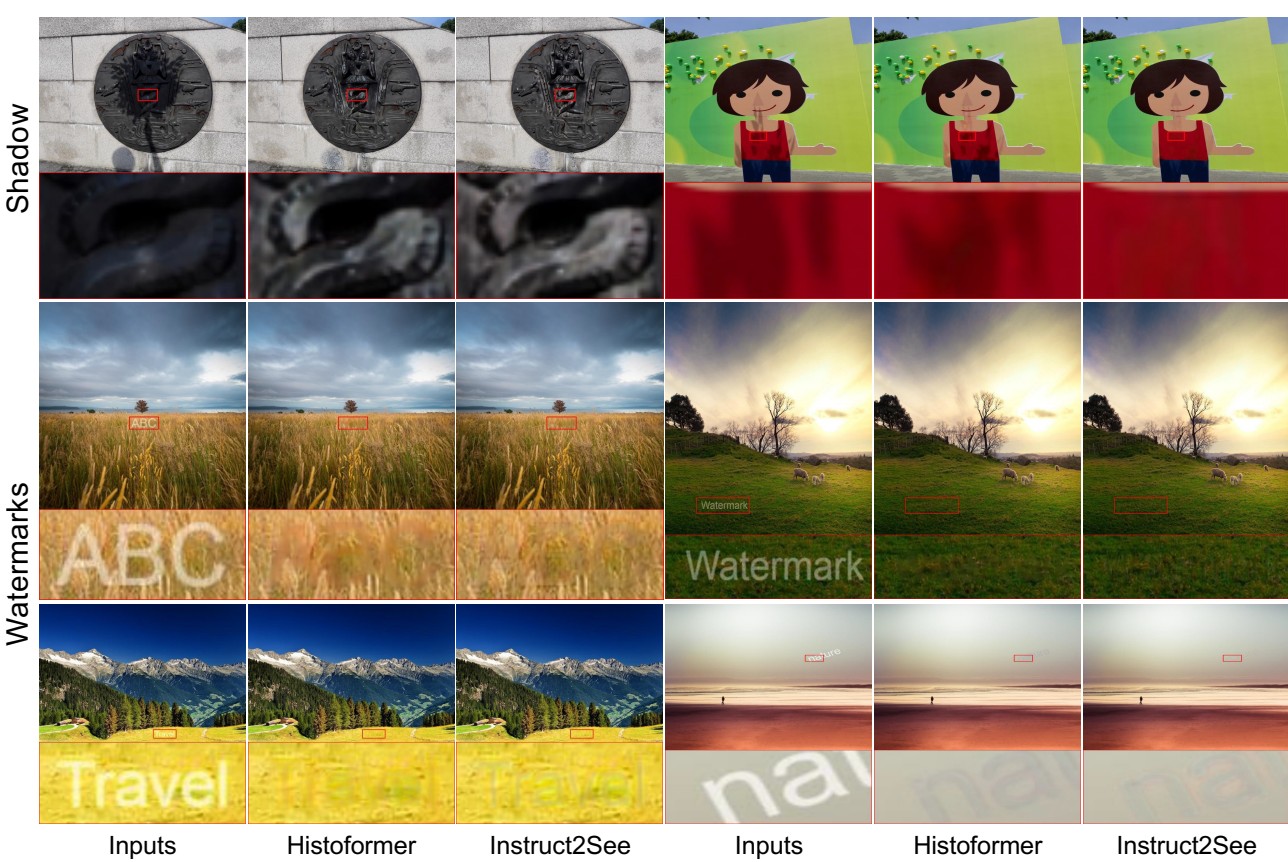

Figure 12: Visual comparisons on more uncommon obstructions.

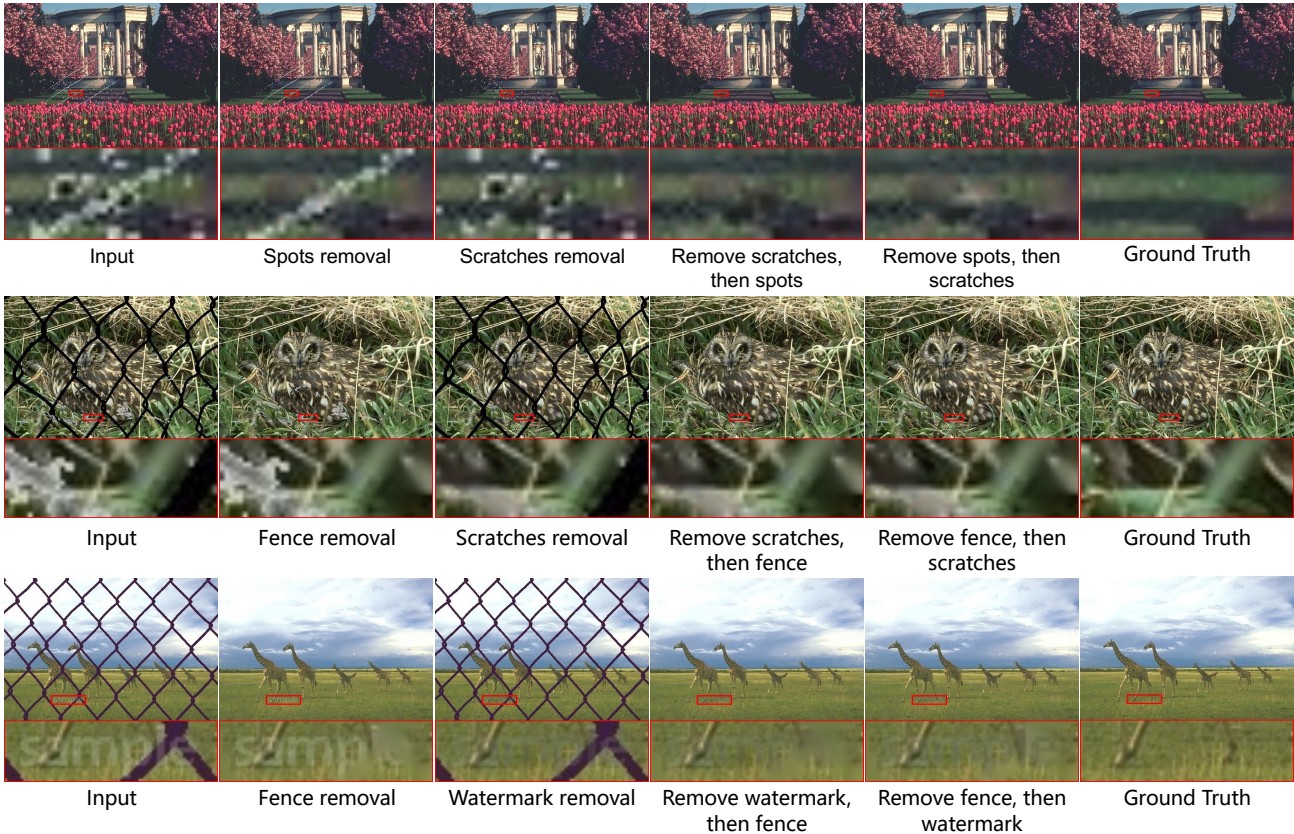

Figure 13: Visual results on multiple obstruction removal.

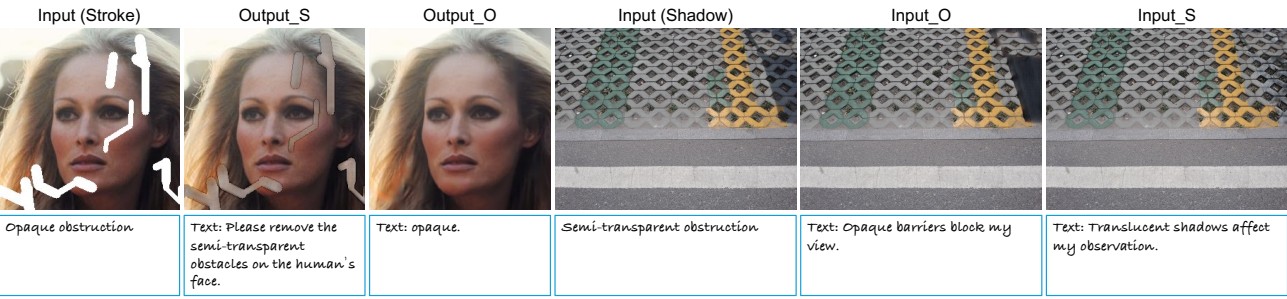

Figure 14: Visual comparisons of using different text descriptions.

