# OpenReview forum: "Instruct2See: Learning to Remove Any Obstructions Across Distributions"
_ICML.cc/2025/Conference — ICML 2025 poster_

### Official Review · Reviewer_zzkC · 2025-03-01

**Overall Recommendation:** 3

**Summary:**

This paper tackles the problem of obstruction removal in images with transformer-based generative models. The key design lies in the modeling of obstructions and also the alignment between types of obstructions and language descriptions (e.g. "rain drops", "fences"). The learned model achieves comparably better results on several benchmarks.

**Claims And Evidence:**

The authors claimed the following contributions:

- **first unified obstruction formulation**: The authors did provide a formulation, but it is a general and broad formulation that previous works also shared.

- **zero-shot paradigm for obstruction removal, incorporating multi-modal prompts**: The authors did include self-switching modules given both visual and textual prompts. The dynamic soft masking strategy seems new to me.

- **comprehensive experiments, strong zero-shot capability**: The unseen experiments did suggest the model is somewhat superior to previous models.

**Essential References Not Discussed:**

There are no obvious missing references.

**Experimental Designs Or Analyses:**

Though the authors achieve comparatively better results against existing methods, the overall improvement seems marginal.

**Methods And Evaluation Criteria:**

The overall design of the model is intuitive, the modeling of soft and hard masking also seems reasonable for the obstruction types at hand. One minor concern is on the depth of exploration over the design of the text-aligned masking strategy as it seems currently it also relates to tasks like inpainting and editing.

**Other Comments Or Suggestions:**

I guess more representative images in the qualitative visualization could be beneficial for understanding the effectiveness of the proposed method.

**Other Strengths And Weaknesses:**

As mentioned earlier, the current soft masking strategy seems to go beyond obstruction removal alone, the authors might want to dive deeper in to this learning framework

**Questions For Authors:**

See the Strength and weakness section.

**Relation To Broader Scientific Literature:**

The designs could similarly be adapted to tasks like generation, editing, etc.

**Theoretical Claims:**

The theoretical claims are intuitive and easy to follow.

---

> ### Author Rebuttal · Authors · 2025-04-01
>
> We sincerely appreciate Reviewer zzKc's valuable feedback. Our responses to the weaknesses and questions are listed below.
>
> **R-W1 Minor performance improvements:** We would like to emphasize that the primary goal of our method is not to maximize performance for any specific obstruction type, but to achieve robust zero-shot generalization across a wide range of unseen obstructions. From this perspective, our approach shows a clear advantage over existing methods, including those that claim to be category-agnostic and those we adapted to operate in zero-shot settings.
>
> Instead of tailoring the model to known obstruction types, our distribution-agnostic formulation treats all obstructions using a unified approach. This applies regardless of their appearance, transparency level, or spatial structure. As a result, our method delivers consistent performance across both seen and unseen categories, something existing methods cannot achieve without retraining or manual tuning.
>
> **R-W2 Selection of more representative cases:** Thank you for your suggestion. We will replace more representative experimental samples for visualization.

---

### Official Review · Reviewer_4Rdn · 2025-03-05

**Overall Recommendation:** 3

**Summary:**

This paper introduces a zero-shot framework for image restoration that can handle a wide range of obstructions, including those not seen during training. Overall, the paper contributes a flexible, distribution-agnostic method for obstruction removal that harnesses multi-modal cues and dynamic masking to achieve robust performance across diverse and unpredictable obstacles.

**Claims And Evidence:**

The submission puts forward several key claims, and overall, many of these claims are supported by a comprehensive set of experiments and analyses.

**Essential References Not Discussed:**

No

**Experimental Designs Or Analyses:**

I check the soundness/validity of any experimental designs or analyses.

**Methods And Evaluation Criteria:**

The contrast experiment and evaluation criteria make sense.

**Other Comments Or Suggestions:**

None.

**Other Strengths And Weaknesses:**

Strength:

1. The paper proposes a unified obstruction removal method by formulating the task as a soft-hard mask restoration problem, effectively integrating visual and textual information, which demonstrates strong theoretical innovation.

2.  It adopts a multi-modal prompting strategy along with an adaptive mask module, enhancing the model’s generalization ability in handling unseen obstructions, with extensive experiments validating its effectiveness.

Weakness:

1. The experiments on unseen obstructions are relatively limited in scope; further expanding the evaluation range may better demonstrate the robustness of its zero-shot learning capability.

2. The technical contribution is somewhat limited; while the multi-modal prompting and mask recovery techniques are effective, they do not substantially deviate from established methodologies, indicating a reliance on existing concepts rather than offering groundbreaking innovations.

**Questions For Authors:**

In Supplementary Material B, we observe that the authors trained CLIP to better adapt to the dynamic soft masking approach. In fact, is it reasonable to train only the text module within the multimodal framework, and could not training the vision encoder affect consistency?  Furthermore, based on our concerns mentioned above, we would like to ask whether the use of a multimodal model is necessary, or if it is possible to use separate modules for visual and textual input.

**Relation To Broader Scientific Literature:**

The key contributions and ideas include:

• Adaptive Masking with a Tunable Adapter: Depending on whether an obstruction has clear (hard) or ambiguous (soft) boundaries, the model dynamically adjusts the mask. This adapter refines the initial mask estimates, enabling more accurate removal, particularly for semi-transparent obstructions like raindrops.

• Zero-Shot Generalization: Extensive experiments show that Instruct2See not only performs well on in-distribution obstructions (like fences, raindrops, and flares) but also generalizes effectively to unseen types of obstructions (e.g., power cables, yarn, scratches).

• Empirical Results: The paper provides comprehensive quantitative and visual comparisons with state-of-the-art methods. The proposed approach often demonstrates improved PSNR/SSIM scores and superior visual quality, especially in challenging, real-world scenarios where traditional models may fail.

**Theoretical Claims:**

I’ve checked the correctness of the proofs in the paper.

---

> ### Author Rebuttal · Authors · 2025-04-01
>
> We sincerely appreciate Reviewer 4Rdn's valuable feedback. Our responses to the weaknesses and questions are listed below.
>
> **R-W1 Limited scope of experimentation:** Thank you for your comment. We believe some of our experimental results may have been inadvertently overlooked. In both the main paper and supplementary materials, we present extensive zero-shot evaluations across a wide range of obstructions, including **rain streaks, snow, strokes, power cables, spots, scratches, yarn, shadows, watermarks**, and **complex multi-obstruction scenarios**. These results collectively validate the effectiveness and generalization capability of our method.
>
> Unlike object removal, where datasets are abundant and well-defined, obstructions are often **irregular in shape, appearance, and distribution**, making comprehensive evaluation more challenging. To address this, we have curated and tested on **all publicly available obstruction data** that aligns with our problem definition. We will revise the manuscript to highlight these results more prominently. Additionally, we welcome suggestions on any other publicly available datasets we may have missed and are happy to include further evaluations as needed.
>
> **R-W2 Limited technical contribution:** Thank you for your feedback. We respectfully believe this concern may arise from an underappreciation of the conceptual novelty behind our approach. Our method introduces a new formulation of obstruction removal as a distribution-agnostic soft masking problem, which fundamentally unifies the treatment of both opaque and semi-transparent obstructions within a single framework.
>
> This perspective shifts away from traditional category-specific pipelines and reframes obstruction removal as a context-aware reconstruction task, where the model learns to reason about partially visible content regardless of obstruction type. Prior methods typically rely on known obstruction categories or retraining per class, whereas our method enables zero-shot generalization to unseen obstruction types—an ability that existing approaches lack.
>
> Moreover, our flexible soft-mask recovery strategy, combined with multi-modal prompt integration, allows the model to adaptively handle obstructions with varying degrees of transparency, shape complexity, and semantic ambiguity. We also show that even with access to ground-truth masks, conventional methods fail to generalize across distributions, underscoring the need for our proposed formulation.
>
> This unified, distribution-agnostic perspective and the demonstrated generalization across diverse and unseen scenarios represent a meaningful advancement in both the theoretical framing and practical capabilities of obstruction removal.
>
> **R-Q1 CLIP fine-tuning strategy:** In our framework, the primary role of the text encoder is not to align text with visual features in the conventional sense, but rather to help the model interpret user instructions, particularly in understanding the transparency attributes of obstructions described in the prompt. To achieve this, fine-tuning the text encoder is essential.
>
> For example, before fine-tuning, the cosine similarity (after softmax) between the user instruction *"There are raindrops in the image, please remove them"* and the two core commands *"remove opaque obstructions"* and *"remove transparent obstructions"* were 0.5374 and 0.4626, respectively. After fine-tuning, these shifted to 0.00004 and 0.99996, respectively, indicating a dramatic improvement in semantic alignment. This refinement is critical for guiding the model’s recovery strategy based on the nature of the obstruction.
>
> Additionally, Section 5.3 (Ablation Study) in the main paper provides experimental evidence for the importance of textual and visual conditioning. We further support this with a qualitative case study in Appendix D.2, illustrating the performance gains achieved under different conditioning strategies.

---

### Official Review · Reviewer_91FV · 2025-03-12

**Overall Recommendation:** 3

**Summary:**

In this paper, the author propose the Instruct2See, which is a zero-shot framework for removing both seen and unseen obstructions from images. It formulates obstruction removal as a soft-hard mask restoration problem, integrating multi-modal prompts via cross-attention. A tunable mask adapter refines masks for semi-transparent obstructions. The results demonstrate the outperforms state-of-the-art methods on PSNR and SSIM while generalizing well to unseen cases.

**Claims And Evidence:**

The paper claims "Remove Any Obstructions." However, I would like to know whether there are any domain restrictions or size limitations.



Other claims I think are clear.

**Essential References Not Discussed:**

No

**Experimental Designs Or Analyses:**

I think the results are promising, and the comparisons are comprehensive. However, I suggest that the author provide more implementation details.

**Methods And Evaluation Criteria:**

I think it makes sense.

However, I would like to understand the computational efficiency. Although Table 6 presents the results, I am still concerned about why the proposed model has the largest number of parameters, yet the FLOPs and runtime are not significantly higher.

**Other Comments Or Suggestions:**

Typo: L802 As shown in...

**Other Strengths And Weaknesses:**

I am concerned about how the model performs in specific domains, such as the medical field.

**Questions For Authors:**

Can the method handle overlapping obstructions (e.g., rain + snow) without interference?

**Relation To Broader Scientific Literature:**

The work builds on vision-language models (CLIP). It advances the field by unifying multi-modal prompts for zero-shot generalization, addressing limitations of task-specific and all-in-one models.

**Theoretical Claims:**

The paper lacks formal theoretical proofs but provides a clear mathematical formulation.

---

> ### Author Rebuttal · Authors · 2025-04-01
>
> We sincerely appreciate Reviewer 91FV's valuable feedback. Our responses to the weaknesses and questions are listed below.
>
> **R-W1 Domain restributions:** Our model is trained on natural image datasets, and as such, it performs well across a wide range of domains within the natural image space, such as urban scenes, landscapes, wildlife, and daily objects, without restriction. However, for domains outside this distribution, such as medical imaging, the model cannot be expected to produce meaningful restorations, as the training data does not include such content.
>
> However, our framework itself is domain-agnostic. By training on appropriately curated domain-specific data (e.g., medical images), the same architecture and methodology can be applied to obstruction removal in those domains. We will clarify this point in the revised manuscript to better reflect the scope and potential of our approach.
>
> **R-W2 Explanation of running efficiency:** Thank you for your question. The efficiency of our method is largely due to the encoder-decoder architecture of the restoration model. Although the model has a relatively high parameter count, this does not translate into high computational cost. This is because deeper modules operate on low-resolution feature maps, allowing for higher channel dimensions without significantly increasing FLOPs or runtime. This design decouples parameter size from computational load, enabling efficient resource use while maintaining strong performance.
>
> **R-W3 More details of experiment:** Thank you for the suggestion. We will revise the Experiment Settings section to include additional implementation and training details for improved clarity and reproducibility.
>
> **R-W4 Minor spelling error:** Thank you for pointing this out. We will carefully review the manuscript to correct any spelling errors.
>
> **R-Q1 Overlapping obstruction removal:** Thank you for your comment. We would like to clarify that our original submission includes experiments on images with multiple overlapping obstructions, with results presented in Appendix C.2 and Figure 11. These experiments demonstrate that our method can accurately identify and represent overlapping obstructions using multimodal prompts and masks, and that our model effectively removes them, further validating the robustness of our approach.

---

### Official Review · Reviewer_17U3 · 2025-03-17

**Overall Recommendation:** 4

**Summary:**

The paper provides a novel pipeline for “obstruction removal”: the combined task of identifying unwanted obstructions in an image and filling in the masked regions with plausible pixels. The method proposes a) using a hybrid adaptable masking strategy to identify the occluders and b) learning a text and vision conditioned model to make the inpainting process “context-aware”. These changes lead to competitive quantitative and qualitative results on seen occluders, and better performance with unseen occluders.

**Claims And Evidence:**

In general, this paper backs up the claims it makes with reasonable evidence. The main claim of the paper is that their method performs on par, and sometimes worse or better than baselines on the task of obstruction removal. This seems to be well justified through standard PSNR / SSIM metrics. Evaluation is split between seen and unseen object classes, and many ways of visualizing the difference (eg. tables, qualitative visualization, comparison plot across baselines eg. Fig 5) are used to justify it. For different parts of the model (use of masking instead of end-to-end, cross attention module for conditioning, and adaptiveness in the masking), an ablation study is provided to show quantitative improvements.

**Essential References Not Discussed:**

I am generally unfamiliar with research in this sub-field and would not know if key baselines were not listed here.

**Experimental Designs Or Analyses:**

Other than the question of how the dataset was chosen and organized, the paper follows standard experimental design and analysis in comparison to a range of baselines, showing many quantitative and qualitative results and using standard metrics in computer vision for image quality. I reviewed all experimental details, including both tables and all details. Assuming that correct datasets were chosen, experiments are sound and clear.

**Methods And Evaluation Criteria:**

At a high level, the method seems to be well motivated and clearly explained. However, key details which are present in the figures (eg. text / image encoders) are not included in writing. There were a lot more details provided in the appendix (a clear algorithm for inference, what actually goes into the text encoder), but this was not clear in the main paper. I believe the methods section needs to be augmented with more details about how corresponding text is generated for the datasets, and how a user will actually interact with the system using prompts.

A quick glance at a close baseline (PromptIR) seems to suggest that there are many other datasets / benchmarks being used in this family of work. As I am not an expert in obstruction removal, it is unclear to me whether any of these datasets are transferable to the current work. I believe the work should augment it's evaluation criteria reading to better describe why particular datasets were chosen, and add a section in the supplementary to describe why datasets used in comparable works were left out.

**Other Comments Or Suggestions:**

As a result of 1) incomplete descriptions of related work, 2) lack of failure analysis/shortcomings, 3) lack of technical description precisely characterizing what kind of “unseen” objects we’d expect this method to work on, and 4) belief that the work would be a stronger fit for a vision rather than a machine learning conference due to lack of ML-direct writing, the present review is only that of weak acceptance. The review is especially borderline in belief.

Given lack of full understanding of benchmarks of this particular sub-field, I am keen to see the other reviews and will be open to change  depending on author / other reviewer’s clarifications.

**Other Strengths And Weaknesses:**

This likely involves a summary of the other sections of the review with a few additional points.

Strengths

For the experiments the paper includes, they are sound and clear to the extent they are described. The paper is full of qualitative visualizations showing test cases in which the method works well, and all baselines are included in these visualizations.

Weaknesses

Writing is missing key details on related work, dataset choice and prompting / text conditioning details of method.

A large part of some critical important parts of the method and results have been put in the appendix / supplementary despite many repeated results being in the main paper. This includes the description of how the CLIP text encoder is finetuned, some failure analysis and comparison to image inpainting methods.

The paper does an insufficient job of limiting the scope of what works with their method, and providing technical motivation for the precise unseen cases that the technical changes should help (or not help) on. For example, after reading the paper, it is clear to me that the method performs better on three sub-classes of unseen obstacles, and some miscellaneous novel obstacles by looking at the qualitative and quantitative results. However, it is not clear why how these unseen classes precisely differ from the seen classes, how large the distribution shift is, or how robust the method will be to other forms of perturbations. It is also clear from the appendix that one failure mode is providing the unintended input text, but there is no analysis presenting the failure mode of the method when it is used as intended. For example, why were the three particular unseen classes chosen to evaluate? And what are some out of distribution unseen classes.

**Questions For Authors:**

1) Why is it that post-masking, an image diffusion model with one of many inpainting / imputation strategies cannot be applied for this task? Such models, since they’re trained at scale will naturally handle a wide range of objects - potentially even in a zero shot way? What prevents numbers from such an approach to be reported alongside other baselines?

2) How was the dataset selection for this method done? What are the standard datasets in this field and why was a subset of BSD chosen for evaluation? Why can’t datasets used by prior work (eg. Prompt IR and Urban100) be also evaluated for this method?

**Relation To Broader Scientific Literature:**

In general, I find the work to be similar to other broad works in vision that use a generative model to solve under-constrained problems in a feed forward manner. The paper will benefit significantly from more description comparing the work in a more detailed manner to other naive approaches (simply using the easiest strategy of inpainting after masking with a foundation diffusion model for image generation) and more problem-specific approaches, while highlighting key differences at both levels. The paper has a very sparse description of related works and differentiation from baselines. While many relevant works seem to be cited, the description don’t clearly highlight the differences in the method in terms of technical contribution. Particularly, I believe revisions are needed to related work to highlight how the key components of the method differ from Restormer, PromptIR, or other close performing baselines. Further,  it is important to characterize the scope in which ease of these baselines work. To this end, many claims in the “Obstruction Removal” section of related work are very general. Lines 083 (“still face challenges”), 085 (“limiting their effectiveness in real world”), etc do not clearly state the differences in failure modes of the two methods.

The writing of the work would benefit from augmenting it to highlight clearly how the model choices enable specific improvements in unseen classes of obstructions. For example, one might expect that conditioning additionally on text should reduce Bayes error of the prediction problem, or using cross attention (instead of another naive conditioning strategy) might enable more precise conditioning on fine-grained features, etc. These will then specifically expand the cases in which the method works compared to baselines. Presently, the writing does not follow this kind of structure clearly motivating what the method enables.

**Theoretical Claims:**

This paper has no theoretical claims.

---

> ### Author Rebuttal · Authors · 2025-04-01
>
> We sincerely appreciate Reviewer 17U3's valuable feedback. Our responses to the weaknesses and questions are listed below.
>
> **R-W1 Lack of key details:** We emphasize that our method targets a fundamentally different problem than prior approaches such as Restormer and PromptIR. These existing restoration methods assume prior knowledge of obstruction characteristics and are designed to handle specific, predefined categories (e.g., rain, snow, flare). As a result, they lack generalization ability: a model trained for snow removal, for example, fails on rain without retraining, even when the distributions are similar.
>
> In contrast, our work breaks this constraint through two key innovations: (1) a distribution-agnostic obstruction formulation, and (2) a flexible soft-mask recovery strategy integrated with multimodal obstruction representation. Together, these enable zero-shot generalization to diverse and unseen obstructions. This direction is both novel and underexplored, marking a significant advancement in the theory and practice of obstruction removal.
>
> We will further strengthen the distinction between our method and existing work in the Introduction and Related Work sections to ensure clarity.
>
> Additionally, we clarify that our original submission already includes: (1) a detailed rationale for dataset selection (see Section 3), and (2) complete details on prompt/text conditions, including their integration in Algorithm 1 (Appendix A) and illustrative examples in Figure 8 (Appendix B). As most of this information is provided in the appendix and may be easily overlooked, we will revise the main text to reorganize the structure or better guide readers to these supporting materials.
>
> **R-W2 Unreasonable distribution between main text and appendices:** Thank you for your suggestion. We will reorganize the content distribution of the main text and appendices.
>
> **R-W3 & Q2 Application scope and problem definition:** Thank you for your feedback. All obstruction categories, both seen and unseen, are sourced from publicly available datasets to ensure reproducibility.
>
> The seen classes (Fences, Raindrops, Flares) were selected for their diversity in shape and structure, providing a strong foundation for generalizing to irregular obstructions. These include grid-like, point-based, and diffuse patterns, offering broad coverage of common obstruction forms.
>
> The unseen classes (Rain Streaks, Snow, Stroke, Power Cables, Yarn, Scratches) were chosen to introduce appearance-level and geometric shifts from the training distribution. For example, snow and yarn differ in opacity and texture; rain streaks and power cables differ in orientation and continuity. This setup enables a meaningful evaluation of zero-shot generalization across varied obstruction types.
>
> We will expand Section 3 to clarify these distinctions and include additional descriptions of seen/unseen differences.
>
> Regarding limitations: Indeed, the limitation of our method under intended usage has already been discussed in Section 6 of the paper. Our approach is specifically designed for small, spatially sparse obstructions and is not suitable for cases where large regions are occluded, as this exceeds the model’s implicit semantic completion capacity. To make this limitation more explicit, we will include additional visual examples and further elaborate on typical failure modes, even within the intended application scope.
>
> **R-Q1 Comparison with diffusion-based method:** Some important comparisons may have been overlooked. As shown in Figure 12 and Table 4 of Appendix C.3, our method demonstrates superior zero-shot generalization compared to inpainting baselines. This includes Repaint, a representative diffusion-based approach. In addition, in response to Reviewer CuLC’s Question W2, we extended our analysis to include DiffEdit, a diffusion-based image editing method. Together, these results provide comprehensive evidence of the advantages of our method over diffusion-based baselines.
>
> | Method       | Rain Streak                      | Snow                             | Stroke                           | Average                          |
> | ------------ | -------------------------------- | -------------------------------- | -------------------------------- | -------------------------------- |
> | LaMa         | 29.07/0.8858                     | 32.32/0.9108                     | 28.10/0.8728                     | 29.83/0.8898                     |
> | RePaint      | 28.78/0.8865                     | 32.20/0.9064                     | 23.78/0.8059                     | 28.25/0.8662                     |
> | DiffEdit     | 23.88/0.6561                     | 24.23/0.6732                     | 11.65/0.6072                     | 19.92/0.6455                     |
> | Instruct2See | **29.82**/**0.8907** | **34.85**/**0.9283** | **29.45**/**0.9067** | **31.37**/**0.9086** |

---

> > ### Comment · Reviewer_17U3 · 2025-04-04
> >
> > Thank you for your response and acknowledging 1) improvements in writing, 2) characterizing distribution shift and 3) point to limitations in scope. To reiterate, I understand the acknowledgement about generalization to larger patches as a limitation - my question was more about limitations on the defined axes that were used to choose the categories (for example: opacity, texture, etc) that experiments and evaluations reflect. To be precise, the paper argue "rain streaks" and "power cables" can be repaired, but it is unclear if a more opaque, or a more high frequency texture could be reconstructed.
> >
> > Thank you for the additional experiment, and clarifying that the approach performs better than DiffEdit. Have these method been trained from scratch on the dataset of choice? My question was more about the importance of pre-trained diffusion models (previous comment: "Such models, since they’re trained at scale will naturally handle a wide range of objects - potentially even in a zero shot way?") and the importance of the proposed design choices when comparing against a  easy-to-use zero shot pre-trained method + editing objective / method (RePaint, DiffEdit with Stable Diffusion) etc, rather than just using a diffusion model instead of a reconstruction objective.
> >
> > Thank you for other clarifications. I will maintain my recommendation for (weak) acceptance for now.

---

> > > ### Author Response · Authors · 2025-04-04
> > >
> > > Thank you for your thoughtful comment. We would like to clarify a few points, as there may still be some misunderstanding regarding our comparisons and experiments.
> > >
> > > Our method is designed to generalize across diverse obstruction types, regardless of transparency or texture. A clear example is the stroke removal task (Figs. 1 and 9), where highly opaque, artificially generated obstructions occlude complex facial features. Our method accurately reconstructs these regions, supporting the strength of our zero-shot formulation.
> > >
> > > Regarding comparisons with Stable Diffusion-based methods, we used official, publicly released models without task-specific fine-tuning to ensure a fair evaluation. All testing was conducted under a strict zero-shot setting, with obstructions differing from training distributions. Results show that while diffusion-based models generalize well in broad tasks, they struggle with precise obstruction removal, where our method performs more robustly.
> > >
> > > We hope this clarifies the design and strengths of our approach. Thank you again for your feedback.

---

### Official Review · Reviewer_CuLC · 2025-03-17

**Overall Recommendation:** 3

**Summary:**

This paper studies obstruction removal of 2D images. The proposed model is a zero-shot method that can handle both seen and unseen obstacles in open vocabulary. The method is to obtain a mask of the obstructions and inpaint/repaint the image with a transformer. The results are claimed to be state-of-the-art.

**Claims And Evidence:**

- Extensive experiments in both qualitative and quantitative manners effectively support the claim of state-of-the-art results.

**Essential References Not Discussed:**

N/A.

**Experimental Designs Or Analyses:**

- The experiment is extensive, and the results can well support the claims.
- The metrics are mainly PSNR and SSIM. It would be better if there were also LPIPS metrics, along with VQAScore, GPTScore, or user studies.

**Methods And Evaluation Criteria:**

- The core of this method contains two parts: (1) a mask detector to predict the obstruction mask, and (2) an inpainting model (that leverages cross-attention, prompts, etc.) to inpaint or repaint the image.
- However, it is completely unclear how this mask detector is designed, constructed, or trained.
    - The word "mask detector" only appears 5 times in the whole paper, merely used as a black box.
    - It is unclear whether the mask detector requires the user to input the type of obstructions or multi-modal clues or only with image input.
    - Given that it is a crucial and integral model component, this becomes a substantial flaw in this paper's technical perspective.
- For the inpainting part, the model itself is straightforward and well-motivated. However, it is also a very typical task for diffusion-based models with the out-of-the-box training-free inpainting method DiffEdit.
    - DiffEdit strictly constraints the modifications within the unmasked part from its implementation, so there will be nothing unrelated changed outside of the mask. The strong capability of diffusion models also have the potential to outperform the proposed method's novel part.
    - I think "applying the predicted mask to Stable diffusion (XL or 3/3.5) + VLM-generated obstruction-free prompt + DiffEdit inpainting" should be considered a naive baseline for comparison.
    - One issue of the naive baseline is some bad support of semi-transparent obstructions as the semi-occluded covered part is unseen to the model. And a very typical way is to follow Instruct-Pix2Pix to channel-wise concatenate the original image in the input, while still inpaint the masked image. If time permits, I would also like to see this results.
- The architecture of the proposed inpainting model is not very clear. It looks like both a VLM that can generate images and a ViT that do dense prediction. I would request more detailed explanation of this architecture, given no code is provided.

**Other Comments Or Suggestions:**

Please refer to the reviews above. Specifically,
- Please provide a detailed description of how the mask detector works.
- Please add the naive diffusion inpainting baseline, at least the training-free one. Note that the prompt can be generated by a VLM, given the image and then removal of all descriptions of the obstruction.

This information is provided in the rebuttal. Therefore, I would like to raise the reviewing score from 2 to 3.

**Other Strengths And Weaknesses:**

- All the figures are small sizes. Even though there is a zoomed-in area, the area itself is too small compared with the full image. Therefore, it is unable to verify whether other parts of the images contain artifacts.

**Questions For Authors:**

Please refer to the reviews above.

**Relation To Broader Scientific Literature:**

This paper proposes a novel method for inpainting in dealing with obstruction removal tasks.

**Theoretical Claims:**

There are no theoretical claims.

---

> ### Author Rebuttal · Authors · 2025-04-01
>
> We sincerely appreciate Reviewer CuLC's valuable feedback. Our responses to the weaknesses and questions are listed below.
>
> **R-W1 Mask detector design:** This concern appears to stem from a misunderstanding. We have already clarified the role and details of the mask detector in Appendix A.1 of the submission. The mask detector is an off-the-shelf component and not central to our contribution. Our primary focus is a novel, distribution-agnostic framework for obstruction removal, which introduces (1) adaptive soft/hard recovery strategies for both opaque and semi-transparent obstructions, and (2) multi-modal prompt integration for precise obstacle representation. The framework is fully plug-and-play and can seamlessly incorporate more advanced mask detectors to further enhance performance without modification.
>
> **R-W2 Comparison with diffusion-based method:** Indeed, our submission demonstrates that diffusion-based models struggle with irregular and complex obstructions. In Table 4 and Figure 12 of Appendix C.3, we compared our method against **Repaint**, a representative diffusion-based inpainting approach, and validated the strong zero-shot generalization ability of our framework through both qualitative and quantitative results.
>
> As requested, we further evaluated **DiffEdit** for unseen obstacle removal (PSNR↑/SSIM↑):
>
>
> | Method | Rain Streak | Snow | Stroke | Average |
> | - | - | - | - | - |
> | DiffEdit     | 23.88/0.6561         | 24.23/0.6732         | 11.65/0.6072         | 19.92/0.6455         |
> | Instruct2See | **29.82**/**0.8907** | **34.85**/**0.9283** | **29.45**/**0.9067** | **31.37**/**0.9086** |
>
> The results show that DiffEdit performs poorly. This stems from its design as an image editing tool, which strictly constrains changes to masked regions and lacks the flexibility to handle irregular-shaped holes. Even when provided with detailed VLM-generated prompts, DiffEdit fails to achieve meaningful completions and often hallucinates entirely new objects. These limitations highlight the inadequacy of diffusion-based editing methods for this task and underscore the effectiveness of our proposed framework.
>
> **R-W3 Usage of input strategy like Instruct-Pix2Pix:** We conducted a direct comparison between our final strategy and the channel-wise concatenation approach (original + masked image), as in Instruct-Pix2Pix. The results are shown in the table below.
>
>
> | Method                 | PSNR↑    | SSIM↑     |
> | - | - | - |
> | Masked Image                  | **30.93** | **0.9250** |
> | Original + Masked Images | 30.19     | 0.9173     |
>
> Incorporating the raw image as input significantly degrades performance rather than improving it. This is because our training data includes three types of degradations, and direct access to the original image encourages the model to memorize specific degradation patterns, which harms zero-shot generalization.
>
> While we agree that conventional inpainting methods struggle with semi-transparent obstructions due to the occluded content being partially visible yet unknown, our method addresses this effectively through a soft-masking strategy. Moreover, original image features are already embedded implicitly via CLIP’s encoder, enabling semantic completion without overfitting to training-specific obstructions.
>
> **R-W4 Description of the restoration model:** The restoration model follows a standard encoder-decoder architecture, a widely adopted design in image processing tasks. As this component is not the core focus of our work (the key is the distribution-agnostic framework), we provided only a brief overview in the main text. However, to ensure methodological transparency and reproducibility, we will include a detailed description of the network architecture in the appendix.
>
> **R-W5 More evaluation metrics:** As suggested, we incorporated three additional evaluation metrics—**LPIPS↓**, **CLIP Score↑**, and **User Study (US)↑** (0–1)—to provide a more comprehensive assessment of obstacle removal performance across both **seen** (Fence, Flare, RainDrop) and **unseen** (Rain Streak, Snow, Stroke) scenarios. Due to space limitations, we reported results for three representative methods. Even under these expanded metrics, our method consistently outperforms baselines. Full comparisons across all metrics and methods will be included in the final submission.
>
>
> | Method       | Seen Obstruction               | Unseen Obstruction             |
> | - | - | - |
> | Restormer    | 0.0746/0.9421/0.74             | 0.1984/0.8833/0.80             |
> | Histoformer  | 0.0798/0.9365/0.84             | 0.1328/0.8885/0.70             |
> | XRestormer   | 0.0924/0.9360/0.68             | 0.1873/0.8820/0.60             |
> | Instruct2See | **0.0694**/**0.9442**/**0.92** | **0.1071**/**0.9140**/**0.84** |
>
> **R-W6 Too small image size:** Thank you for your suggestion. We will modify the experimental images to enhance the comparability between methods.

---

### Decision · Program_Chairs · 2025-05-01

**Decision:**

Accept (poster)

**Comment:**

All reviewers found the proposed method to be effective and the results promising. The rebuttal successfully addressed reviewer comments and all reviewers recommend acceptance. The authors are encouraged to strengthen the final version by incorporating reviewer suggestions, particularly by integrating additional results, clarifying technical contributions beyond prior work, and ensuring key content currently in the appendix is moved to the main paper with improved linkage between the two.

Reviewer CuLC also raised two remaining concerns to be addressed in the revision: (1) while the mask detector is an off-the-shelf component and not a claimed contribution, the paper lacks analysis on how its selection affects performance; (2) although various types of obstructions were evaluated, they share similar characteristics, which may limit generalizability. These issues should be acknowledged and discussed.